# 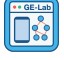 GUI Exploration Lab: Enhancing Screen Navigation in Agents via Multi-Turn Reinforcement Learning

**Haolong Yan**[1*]   **Yeqing Shen**[2*]   **Xin Huang**[3]   **Jia Wang**[2]   **Kaijun Tan**[2]
**Zhixuan Liang**[3]   **Hongxin Li**[4]   **Zheng Ge**[2]   **Osamu Yoshie**[3]
**Si Li**[1†]   **Xiangyu Zhang**[2]   **Daxin Jiang**[2]

[1]Beijing University of Posts and Telecommunications
[2]StepFun   [3]Waseda University
[4]Institute of Automation, Chinese Academy of Sciences

## Abstract

With the rapid development of Large Vision Language Models, the focus of Graphical User Interface (GUI) agent tasks shifts from single-screen tasks to complex screen navigation challenges. However, real-world GUI environments, such as PC software and mobile Apps, are often complex and proprietary, making it difficult to obtain the comprehensive environment information needed for agent training and evaluation. This limitation hinders systematic investigation and benchmarking of agent navigation capabilities. To address this limitation, we introduce GUI Exploration Lab, a simulation environment engine for GUI agent navigation research that enables flexible definition and composition of screens, icons, and navigation graphs, while providing full access to environment information for comprehensive agent training and evaluation. Through extensive experiments, we find that supervised fine-tuning enables effective memorization of fundamental knowledge, serving as a crucial foundation for subsequent training. Building on this, single-turn reinforcement learning further enhances generalization to unseen scenarios. Finally, multi-turn reinforcement learning encourages the development of exploration strategies through interactive trial and error, leading to further improvements in screen navigation performance. We validate our methods on both static and interactive benchmarks, demonstrating that our findings generalize effectively to real-world scenarios. These findings demonstrate the advantages of reinforcement learning approaches in GUI navigation and offer practical guidance for building more capable and generalizable GUI agents.

## 1   Introduction

Large Vision-Language Models (LVLMs) [2, 29, 1, 38] drives significant advances in the development of GUI agents [31, 44, 28]. At a high level, GUI agent tasks [35, 34, 12, 45, 46] fall into two main categories: single-screen task and screen navigation. For example, the task "Order an iPad from the Apple website with GUI-Agent engraving" can be decomposed into subtasks such as single-screen task (e.g., entering the required information to place an order) and screen navigation (e.g., navigating to the iPad purchase page). As the grounding capabilities of current mainstream LVLMs [13, 43, 31, 28, 18] continue to improve, tasks such as form completion and target icon selection no

---

*Equal contribution.
†Corresponding author.

39th Conference on Neural Information Processing Systems (NeurIPS 2025).

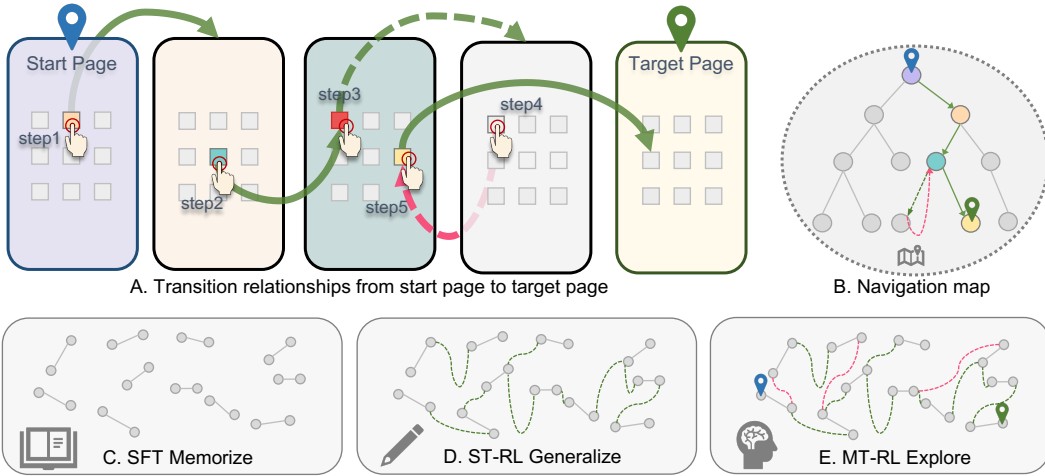

Figure 1: (A) Page transition sequences: green arrows show optimal paths, red arrows show detours. (B) Tree-structured navigation map with optimal (green) and redundant (red) paths. (C) SFT enables memorizing page transitions for basic navigation. (D) ST-RL generalizes to unseen paths, enhancing navigation. (E) MT-RL improves exploration and task success through environment interaction.

longer pose a major obstacle to task completion. Instead, the main challenge for GUI agents now lies in navigating complex screens [31], which requires understanding the environment's navigation map and clicking the correct icons to reach the target page.

To enhance the screen navigation capabilities of GUI agents, it is crucial to have an environment that provides accurate screen layout information and inter-screen navigation graphs for training and evaluation. However, real-world GUI environments, such as PC software and mobile Apps, are often complex and proprietary, making it difficult to obtain the comprehensive environment information needed for these purposes [12, 46, 45]. This limitation hinders systematic investigation and benchmarking of agent navigation capabilities. To address this, we introduce GUI-Exploration Lab (GE-Lab), a simulation environment engine specifically designed for advancing GUI agent navigation, in which screens, icons, and inter-screen navigation graphs can be flexibly defined, while providing full access to environment information for comprehensive training and evaluation.

Within the GE-Lab environment, as illustrated in Figure 1, it is straightforward to obtain both single-step transition knowledge, such as navigating from the current screen node to a new node by clicking a specific icon, and multi-step transition knowledge, which involves reaching any target screen node from the current node through a sequence of actions (see Figure 1A, B). Existing approaches based on SFT require large volumes of data and heavily depend on expert trajectories [34, 20, 5, 13, 31]. However, collecting high-quality trajectory data on real-world GUI platforms is costly [12], and it is challenging to assess the coverage of collected trajectories relative to the full set of possible transitions available on the platform. In contrast, the GE-Lab environment enables low-cost scaling of trajectory data collection and facilitates systematic ablation studies. As shown in Figure 1C, SFT provides a crucial foundation for screen navigation, suggesting that sufficiently large trajectory datasets can enable accurate navigation between arbitrary screens. In Figure 1C, only nodes with degree 1 represent training data that contains knowledge about two connected nodes. This does not imply that every node in the actual state transition graph has a degree of 1.

Nevertheless, SFT exhibits limited generalization to unseen environments [9, 44, 28, 27, 5]. Recent ST-RL methods, such as UI-TARS [31] and UI-R1 [28], leverage preference optimization and rule-based rewards to improve generalization. In the GE-Lab simulation environment, access to each screen's layout and navigation graph enables low-cost experimentation with various ST-RL methods. As shown in Figure 1D, these methods demonstrate better generalization than SFT. ST-RL methods focus on correcting single-step actions, while real-world interactions often involve longer action chains. MT-RL, inspired by RAGEN [42] and VAGEN [40], offers advantages for training agents through interaction. GE-Lab provides the first GUI simulation engine for MT-RL with interactive training and accurate reward signals, addressing practical RL challenges through curriculum learning

and data proportion adjustment. As shown in Figure 1E, MT-RL encourages exploration and further improves task success rates. Figure 1A, B illustrate the agent's navigation process, including backtracking and successful task completion. Moreover, MT-RL reduces dependence on annotated data by relying on instructions and interactive environments with reward signals. Moreover, we validate our methods on both static and interactive benchmarks, demonstrating that our findings generalize effectively to real-world scenarios.

To summarize, our key contributions are the following: (1) We present GE-Lab, a simulation environment engine that flexibly defines screens, icons, and navigation graphs, providing full access to environment information for comprehensive GUI agent training and evaluation. (2) Extensive experiments in GE-Lab reveal a clear training paradigm: supervised fine-tuning memorizes fundamental navigation knowledge, ST-RL enhances generalization to unseen scenarios, and MT-RL promotes exploration and further boosts navigation performance while reducing reliance on annotated data. (3) Our systematic benchmarking clarifies the strengths of reinforcement learning in GUI navigation and offers concrete guidance for building more generalizable agents.

## 2 Related work

### 2.1 GUI Agents

In recent years, GUI agents experience significant evolution. Early methodologies predominantly rely on HTML structures and accessibility trees [23, 11, 52, 25], yet these approaches encounter substantial cross-platform compatibility challenges due to potential information gaps across diverse software environments. The advent of LVLMs marks a pivotal shift in this domain [14, 43, 8], facilitating the integration of GUI comprehension and device control within unified frameworks. Current research in LVLM-powered GUI agents follows two primary technical trajectories: one leverages closed-source models with advanced prompt engineering [48, 21, 39], while the other enhances open-source models through fine-tuning for GUI interaction [34, 20, 5, 13]. Existing agents [13, 43, 31, 28, 18] exhibit robust visual grounding capabilities, effectively supporting basic operations such as target icon selection. However, the principal challenge lies not in executing individual clicks but in navigating complex screen transitions based on task objectives. Addressing this navigation challenge, UI-TARS [31] extended fundamental grounding capabilities by introducing several GUI-specific competencies, including element description, GUI dense captioning, and state transition captioning, while incorporating task decomposition and trial-and-error strategies to learn inter-screen transition. Nevertheless, SFT necessitates a large volume of manually annotated real GUI trajectory data [34, 20, 5, 13, 31] and exhibits limited generalization to unseen environments [9, 44, 28, 27, 5]. Consequently, recent research on agents increasingly shifts towards RL approaches.

To evaluate GUI agent performance, two primary categories of benchmarks are developed: static benchmarks, including ScreenSpot [8], ScreenSpot-v2 [43], FuncPred [19], MoTIF [4], Refexp [33], Llamatouch [50], VWB AG [24], and VWB EG [24], primarily assess task execution correctness through ground truth matching to evaluate grounding and planning accuracy; and interactive benchmarks AndriodWorld [35] and OSWorld [45], which enable agents to execute tasks within virtual machines and determine task success through components that evaluate actual task completion.

### 2.2 Reinforcement Learning

RL is a powerful paradigm for training agents to make optimal decisions through interactions with environment [3, 10], enabling them to autonomously learn complex behaviors by maximizing cumulative rewards. UI-TARS [31] employs Direct Preference Optimization (DPO) [32], leveraging incorrect rollout actions paired with human-corrected actions for error correction. DeepSeek-R1 [36] demonstrates the efficacy of rule-based RL for mathematical problem-solving. Subsequently, [26, 41, 30, 15, 6, 51, 7, 37] extend this framework to RL, achieving significant improvements in vision-related tasks such as visual grounding. UI-R1 [28] and GUI-R1 [44] utilize Group Relative Policy Optimization (GRPO) [36], training with model rollouts and rule-based rewards guided by ground truth. However, these methods focus solely on single-step action correction, whereas real-world interactions are often more complex and involve longer action chains. Outside the GUI domain, several innovative agent systems [17, 16, 22, 42, 40] leverage interactive environments to train agents through reinforcement learning, thus enhancing their decision-making and execution capabilities. Notably, RAGEN [42] and VAGEN [40] leverage MT-RL strategies to effectively enhance agents'

capabilities in environmental interaction and multi-step reasoning. These developments inspire our approach of combining MT-RL with the GE-Lab environment to establish robust screen navigation capabilities for GUI agents, addressing a critical challenge in real-world GUI environments.

# 3 Method

## 3.1 GUI Exploration Lab

To address the challenges posed by complex engineering issues in real-world GUI platforms that make them unsuitable for interactive environment training, we propose GE-Lab, a novel simulation environment engine that generates diverse GUI environments for flexible agent training and evaluation. As illustrated in Figure 1B, we model the simulated environment as a tree structure where nodes represent screens and edges represent clickable transitions between screens. Both the number of nodes and the connectivity of edges within the environment are fully customizable. Specifically, GE-Lab first generates a graph of the environment based on user-defined parameters, which determines the navigation relationships between different screens. Then, GE-Lab renders a screen for each node of the graph, with each screen displaying n icons (where n is the degree of the node), where clicking an icon triggers a screen transition. The icons appearing on each screen are collected from the Internet and randomly sampled from an icon library, intentionally selected to be uncommon in typical applications and randomly named to prevent the model from leveraging prior knowledge about icon semantics, thereby enhancing the diversity of the simulation environment. During environment construction, both the selection and placement of icons are randomized, facilitating robust OOD testing. In addition to these randomly assigned functional icons, each screen also contains "home" and "back" icons, enabling navigation to the root and parent nodes.

During training, the metadata generated by GE-Lab allows for the construction of screen navigation tasks. For instance, at the page_0 node, a task instruction such as "From page_0 to page_6" can be generated. Leveraging the navigation map, it is easy to synthesize trajectory data between arbitrary screens at low cost, supporting both shortest-path and redundant-path trajectories. Furthermore, agents can be trained interactively: the agent receives a image of the current screen and outputs an action, the environment executes the action to transition to a new state, and the environment can accurately determine whether the agent reaches the target screen, providing precise rewards. During evaluation, the simulation environment readily supports both static benchmarks and end-to-end interactive assessments.

## 3.2 Partially Observable Markov Decision Process Framework

### 3.2.1 Problem Formulation

The operation of a GUI Agent involves step-by-step interaction with an environment. The GUI agent operates under conditions of partial observability, wherein it lacks access to the complete environmental state and must instead identify interactive regions based solely on the current screen state. Through the integration of historical states with present observations to inform decision-making processes, this navigation paradigm is appropriately formalized within the POMDP framework, defined by the components $(\mathcal{U}, \mathcal{A}, \mathcal{S}, \mathcal{O}, T)$, as shown in Figure 2A,C. These components correspond to the task space $\mathcal{U}$, action space $\mathcal{A}$, state space $\mathcal{S}$, and observation space $\mathcal{O}$. The probabilistic transitions between states are governed by $T : \mathcal{S} \times \mathcal{A} \rightarrow P(\mathcal{S})$, which maps a state and action to a distribution over the next possible states.

Functionally, the GUI agent acts as a policy $\pi$, mapping observations to actions $\pi : \mathcal{O} \rightarrow \mathcal{A}$. Given a specific task $u \in \mathcal{U}$, the agent executes a series of actions aiming for completion. At each time step t, using the current observation $o \in \mathcal{O}$, the agent's policy $\pi$ selects the next action $a \in \mathcal{A}$. The environment then evolves to a subsequent state $s' \in \mathcal{S}$ according to the transition model $T$. Optionally, the agent may receive a reward $r = R(s, a, s')$, which is calculated based on the reward function $R : \mathcal{S} \times \mathcal{A} \times \mathcal{S} \rightarrow \mathbb{R}$.

### 3.2.2 Agent Protocol

Our agent receives three primary input components: task instruction, history, and current state. The task instruction $u$ defines navigation objectives such as "From page_221 to page_151". The

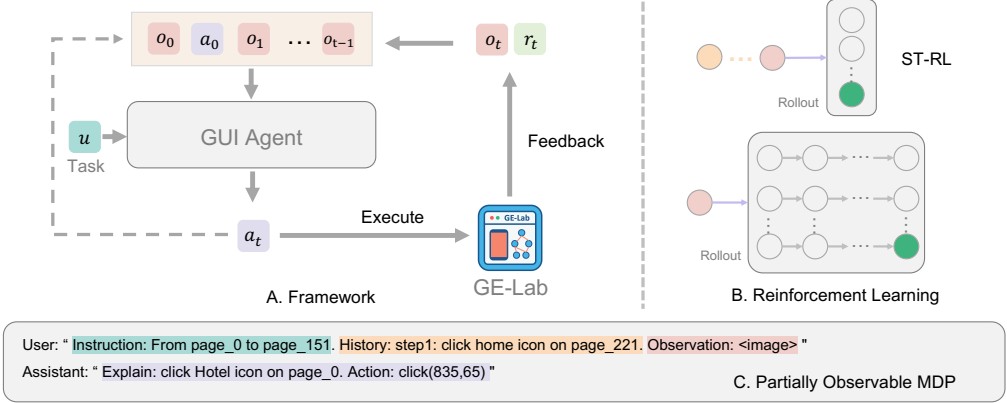

Figure 2: (A) Overview of the interactive training framework: the GUI agent receives observations ($o$), actions ($a$), rewards ($r$), and interacts with the GE-Lab environment to receive feedback and execute actions. (B) Reinforcement learning: ST-RL operates on pre-constructed trajectories, while MT-RL extends to multi-step online rollouts, enabling the agent to generate observation-action sequences interactively within the environment. (C) The navigation task is formalized as a POMDP, where historical states and current observations jointly inform the agent's decisions.

history component maintains a record of previous observations $o$ and actions $a$ (e.g., "step1: click the home icon on page_221"), providing temporal context for sequential decision making. The current observation is a screenshot image of the GE-Lab environment, showing the unfamiliar GUI icons with which the agent can interact. The GUI agent attends to task instructions at each step, makes decisions informed by textual representations of historical states and actions, and generates appropriate actions based on the current screen state. This approach effectively mitigates task forgetting, reduces action repetition errors, and prevents state interference that would otherwise arise from maintaining excessively long sequence lengths containing complete historical screen states.

For action generation, our framework adopts ReAct [47] to formulate appropriate actions within the GE-Lab environment. This approach integrates reasoning and action generation by requiring the agent to produce an explanation of its intended operation along with a precise action command (e.g., "click Home icon on page_0; Action: click(635,65)"), which can be directly executed in the GE-Lab environment. In both MT-RL training and inference, we set a maximum number of interaction rounds (e.g., 12). According to the agent protocol, with each image encoded as 729 tokens, the current maximum context token length, which is the total of the instruction, history, and observation, is less than $1k$ tokens.

### 3.3 Reinforcement Learning

#### 3.3.1 Single-Turn Reinforcement Learning

The ST-RL paradigm, based on the GRPO [36], addresses the screen navigation challenge in partially observable GUI environments (as shown in Figure 2B). At each time step $t$, the agent observes $\bar{o}_t \in \mathcal{O}$ and receives a task specification $u \in \mathcal{U}$. The agent selects an action $a_t \in \mathcal{A}$ to interact with the environment, where the underlying state $s_t \in \mathcal{S}$ is partially observable. The policy $\pi_\theta$ is thus conditioned on the current observation, the history of past observations and actions, and the task:

$$\pi_\theta(a_t|\bar{o}_{0:t}, \bar{a}_{0:t-1}, u), \tag{1}$$

where $\bar{o}_{0:t}$ and $\bar{a}_{0:t-1}$ denote the sequences of observations and actions up to time $t$ (the overline $^-$ indicates pre-constructed trajectories), and $a_t$ is the action generated by the model at the current step via rollout. For ST-RL, $\bar{o}_{0:t}$ and $\bar{a}_{0:t-1}$ are pre-constructed trajectories, which correspond to the orange history part in Figure 2, while $a_t$ is the action generated by the model at the current step. The objective is to maximize the single-step return:

$$J(\theta) = \mathbb{E}_{\bar{o}_{0:t}, \bar{a}_{0:t-1}, a_t} \left[ r(\bar{o}_t, a_t, u) \right], \tag{2}$$

where $r(\cdot)$ is a composite reward that provides explicit supervision for GUI interaction, enhancing both training efficiency and navigation performance. Specifically, the reward consists of four

Table 1: Models Performance on In-Distribution, Out-of-Distribution, and Interactive Benchmarks

| Model | ID | | | OOD | | | Interactive | |
|---|---|---|---|---|---|---|---|---|
| | Edge | Path | Overall | Edge | Path | Overall | Pass@1 | Pass@5 |
| Non-fine-tuned Model | | | | | | | | |
| GPT-4o-2024-11-20 [29] | - | - | - | 34.10 | 5.03 | 25.85 | 1.74 | 2.49 |
| Claude-3.7-Sonnet [1] | - | - | - | 21.77 | 1.92 | 21.52 | 0.43 | 0.61 |
| Gemini-2.0-flash-thinking [38] | - | - | - | 15.05 | 5.33 | 8.80 | 0.36 | 0.52 |
| Fine-tuned Model | | | | | | | | |
| Qwen2.5-VL-7B-SFT | 94.82 | **99.76** | **98.89** | 64.55 | 41.76 | 55.45 | 14.30 | 20.86 |
| Qwen2.5-VL-7B-ST-RL | **97.48** | 97.08 | 97.63 | 68.68 | 52.25 | 63.06 | 17.22 | 22.34 |
| Qwen2.5-VL-7B-MT-RL | 72.60 | 57.77 | 67.33 | **69.86** | **52.35** | **63.25** | **17.47** | **25.16** |

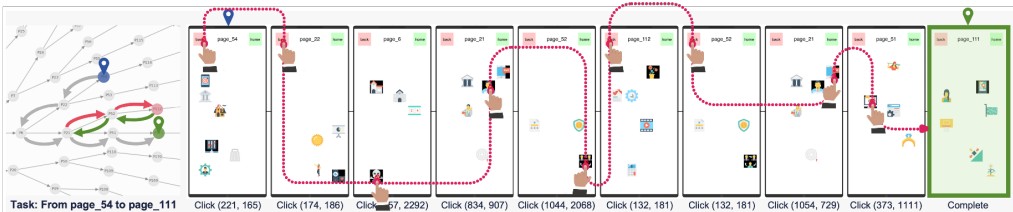

Figure 3: Case Study. Left: Environmental navigation map. Right: Detailed visualization of the page transition flow from the initial to the final state.

components: (1) *Action Type Reward*—evaluates the correctness of the action type; (2) *Coordinate Accuracy Reward*—measures whether the predicted click coordinates fall within the target region; (3) *Intent Matching Reward*—assesses if the selected icon matches the intended interaction; (4) *Format Reward*—validates whether the generated action conforms to the required output format. The reward function formula is provided in Appendix A.4.

### 3.3.2 Multi-Turn Reinforcement Learning

The MT-RL paradigm extends the agent-environment interaction to multiple decision steps, as illustrated in Figure 2B. Unlike ST-RL, where the observation and action sequences $\bar{o}_{0:t}$ and $\bar{a}_{0:t-1}$ are pre-constructed, MT-RL generates these sequences online through interactive rollouts, i.e., $\{o_k, a_k\}_{k=0}^{t} \sim \mathcal{E}(\pi_\theta)$, where $\mathcal{E}(\pi_\theta)$ denotes the environment trajectory induced by policy $\pi_\theta$. At each time step $t$, the agent observes $o_t \in \mathcal{O}$ and selects an action $a_t \in \mathcal{A}$ according to its policy:

$$a_t \sim \pi_\theta(\cdot \mid o_{0:t}, a_{0:t-1}, u), \tag{3}$$

where $u \in \mathcal{U}$ denotes the task specification, and the sequences $(o_{0:t}, a_{0:t-1})$ are dynamically generated during the agent's interaction with the environment. The objective of MT-RL is to maximize the expected final-step reward over the trajectory:

$$J(\theta) = \mathbb{E}_{o_{0:t}, a_{0:t} \sim \pi_\theta} \left[ r(o_t, a_t, u) \right], \tag{4}$$

where the reward function is defined in a sparse, goal-based *A2B* reward: the agent receives a reward of $+1$ if it successfully reaches the target screen node, and $0$ otherwise. This sparse reward formulation encourages the agent to efficiently navigate through the environment to accomplish the specified task. Notably, by providing reward only at the target state, this scheme promotes uninhibited exploration and enables the agent to learn a more comprehensive value function over the state space, facilitating the discovery of diverse and efficient trajectories for task completion.

## 4 Experiment

### 4.1 Environment Configuration

All training is conducted in Env-Base, a tree-structured graph with maximum depth of 7. Excluding system-defined edges representing the home and back, the environment's branching structure follows a node distribution of [5,3,2,2,1,1,0] at each level respectively. In Section 5.2, the Environment

Table 2: Performance Comparison of Methods across Tasks of Varying Difficulty

|        |       | Path@1 | Path@2 | Path@3 | Path@4 | Path@5 | Path@6 | Path@7 |
|--------|-------|--------|--------|--------|--------|--------|--------|--------|
| Pass@1 | SFT   | 99.71  | 51.16  | 19.55  | 8.52   | 3.13   | 2.15   | 0.31   |
|        | ST-RL | 99.71  | 59.73  | 27.57  | 14.01  | 4.59   | 3.38   | 0.83   |
|        | MT-RL | 98.10  | 52.93  | 26.31  | 13.64  | 6.63   | 4.17   | 2.92   |
| Pass@5 | SFT   | 100.00 | 74.15  | 36.04  | 19.75  | 6.71   | 5.04   | 1.30   |
|        | ST-RL | 100.00 | 70.07  | 37.84  | 23.77  | 7.52   | 7.02   | 3.39   |
|        | MT-RL | 100.00 | 66.67  | 43.24  | 24.69  | 13.01  | 8.11   | 8.33   |

Table 3: Performance Comparison of Methods across Out-of-Distribution Environments

|              | SFT     |       |       | ST-RL   |       |       | MT-RL   |       |       |
|--------------|---------|-------|-------|---------|-------|-------|---------|-------|-------|
|              | Overall | Path  | Edge  | Overall | Path  | Edge  | Overall | Path  | Edge  |
| Env-Base     | 55.46   | 41.76 | 64.55 | 63.06   | 52.25 | 68.68 | 63.25   | 52.35 | 69.86 |
| Env-Image    | 32.91   | 7.38  | 63.90 | 33.99   | 7.42  | 69.68 | 34.17   | 7.54  | 70.64 |
| Env-Name     | 32.01   | 6.77  | 61.24 | 32.13   | 5.95  | 64.77 | 32.74   | 6.65  | 65.90 |
| Env-Position | 36.19   | 12.22 | 64.55 | 40.31   | 17.34 | 68.81 | 42.79   | 20.72 | 70.51 |
| Env-Noise    | 41.18   | 21.87 | 57.50 | 42.33   | 23.55 | 57.90 | 44.56   | 27.05 | 60.55 |

Variants are utilized for OOD testing. Specifically, Env-Image, Env-Name, and Env-Position denote modifications to the icon image, icon name, and icon position within Env-Base, respectively. Env-Noise introduces noise icons into Env-Base.

*Edge* traversal tasks represent single-step transitions, while *Path* traversal tasks encompass multi-step navigation sequences. The current environment root node comprises five subtrees: two are designated for SFT training, two for RL training, and one for OOD testing. Notably, the SFT training dataset also incorporates *Edge* data from all subtrees, including the Test subtree. This inclusion is intended to provide the agent with fundamental knowledge of the environment, which can be leveraged for path planning and to prevent the model from making entirely random decisions. In addition, the SFT training set is augmented with grounding and understanding data for the icons used in the Env-Base, further enhancing the agent's ability to interpret and interact with GUI elements. We employ Qwen2.5-VL-7B-Instruct [2] as our foundation model for all experiments.

## 4.2 ST-RL Generalizes on Out-of-Distribution Tasks

Our experimental results, shown in Table 1, provide compelling evidence regarding the effectiveness of different model training strategies for page navigation tasks. Analysis of performance metrics across ID and OOD benchmarks reveals several significant findings. The non-fine-tuned models demonstrate poor performance across all evaluation metrics, with scores approximating random selection levels. This substantiates our hypothesis that general-purpose models without domain-specific training lack the requisite knowledge representations for effective navigation within simulated environments. In contrast, the Qwen2.5-VL-7B-SFT exhibites remarkable improvements, particularly on ID benchmarks. This dramatic performance enhancement indicates the model's strong capacity to memorize domain-specific navigation patterns. However, performance declines substantially on OOD benchmarks, suggesting limited generalization capability when confronted with novel scenarios outside the training distribution. Notably, the ST-RL approach (Qwen2.5-VL-7B-ST-RL) demonstrates superior performance characteristics. While maintaining excellent ID benchmark results, it achieves significantly better generalization on OOD tasks (Overall: 63.06 vs. SFT's 55.45). This 13.7% relative improvement in OOD performance validates our reinforcement learning methodology, which leverages trajectory-dependent reward functions to better capture underlying navigation principles rather than merely memorizing specific action sequences. Furthermore, Interactive Benchmark results reinforce these findings, with ST-RL achieving superior Pass@1 and Pass@5 scores (17.22 and 22.34, respectively) compared to SFT (14.30 and 20.86). This consistent performance advantage across both automated metrics and interactive human evaluation confirms that while SFT effectively memorizes in-domain knowledge, the ST-RL approach yields substantially better generalization to novel scenarios, highlighting the advantages of RL for navigation tasks in dynamic environments.

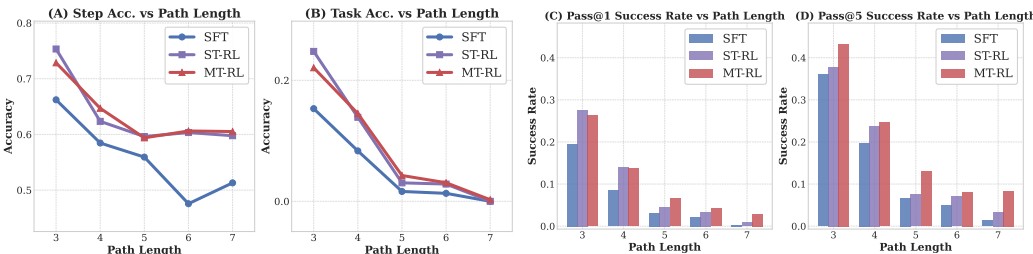

Figure 4: (A-B) Accuracy across different settings and difficulty levels within the static benchmark. (C-D) Success rates across Pass@N and difficulty levels within interactive Benchmark.

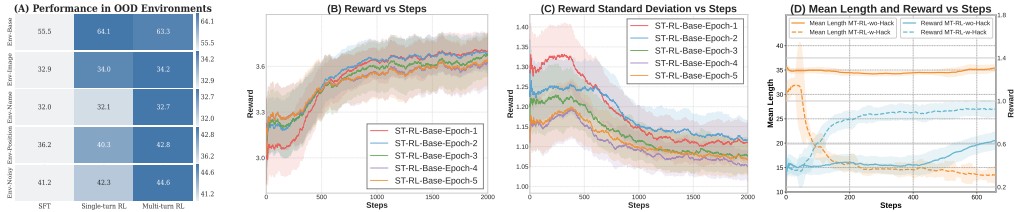

Figure 5: (A) Heatmap visualization of performance in different OOD environments. (B-C) The impact of different SFT stages on ST-RL, showing performance (e.g., mean and standard deviation) respectively. (D) Results of action space modification to counter reward hacking in MT-RL.

## 4.3 MT-RL Explores in Interactive Environments

According to the results presented in Table 1, MT-RL presents a distinctive performance profile across evaluation metrics. Unlike SFT and ST-RL which train on pre-synthesized trajectories, MT-RL employs interactive multi-round training, resulting in lower ID benchmark scores. This indicates MT-RL avoids overfitting to training distributions, instead developing more generalizable navigation strategies. On OOD benchmarks, MT-RL demonstrates comparable performance to ST-RL (Overall: 63.25 vs. 63.06), both significantly outperforming SFT (55.45). The most compelling evidence of MT-RL's effectiveness emerges in Interactive Benchmark, where it achieves the highest scores (Pass@1: 17.47, Pass@5: 25.16). This superior performance in realistic interaction scenarios can be attributed to MT-RL's training paradigm, which aligns closely with actual deployment conditions. By learning through environmental exploration rather than static demonstrations, MT-RL develops robust adaptive behaviors that translate effectively to real-world navigation challenges, enabling the agent to strategically explore and recover from suboptimal states. Figure 3 illustrates our agent's capability to navigate complex screens, demonstrating strategic exploration and error recovery when deviating from the optimal path between page_54 and page_111. After encountering a suboptimal state following the fourth click (834,907), the agent successfully explores alternative paths and backtracks when necessary, ultimately completing the task through multiple recovery steps.

## 5 Analysis

### 5.1 Performance Comparison on Tasks of Varying Difficulty

We conduct ablation experiments to evaluate the performance of our methods across varying task complexities. Figure 4(A-B) shows the step and task success rates in static benchmarks, while Table 2 and Figure 4(C-D) presents Pass@1 and Pass@5 success rates in interactive settings. Across all scenarios, performance drops drastically as path length increases, confirming path length as a key indicator of task difficulty. In static evaluations, ST-RL and MT-RL consistently outperform SFT, especially for longer paths (5–7 steps). This suggests reinforcement learning improves generalization in complex tasks. On interactive benchmark, MT-RL shows even greater gains, beating other methods in Pass@1 and Pass@5 success rates. These results highlight MT-RL's alignment with real-world conditions, where iterative feedback and exploration allow agents to learn from failures and continually refine their strategies, leading to higher success rates.

## 5.2 Generalization in Different Out-of-Distribution Scenarios

We evaluate different approaches in various OOD environments to compare their robustness. Table 3 shows the detailed experimental results. Across all OOD settings, reinforcement learning–based methods (ST-RL and MT-RL) consistently outperform SFT. Notably, in the Env-Base setting, MT-RL and ST-RL achieve success rates of around 63%, compared to 55% for SFT, underscoring better generalization in unseen scenarios. Altering icons or labels (Env-Image and Env-Name) causes significant performance drops (from around 63% to 32–34%; Path success dips to about 7%), highlighting how foundational knowledge about environment elements is crucial for navigation. In contrast, changing icon positions (Env-Position) or adding noise icons (Env-Noise) yields more moderate declines, and MT-RL still maintains higher rates (42–44% overall success, 20–27% on the Path metric), indicating its robustness in dynamic settings. The heatmap in Figure5A confirms these trends, showing MT-RL consistently outperforms ST-RL and SFT across all perturbations.

## 5.3 Influence of SFT on the ST-RL

To investigate the influence of SFT on the subsequent RL training phase, we first subject the base model to SFT for varying numbers of epochs. Subsequently, models from different training stages are utilized to initialize the ST-RL training. Figure 5B presents the reward and training steps for ST-RL training initiated with models that have undergone SFT for 1 to 5 epochs. It is clearly observable that models with different degrees of SFT training exhibit distinct learning trajectories and ultimately achievable reward levels during the subsequent RL training. A core finding is that RL training initialized with models from early SFT stages achieves higher cumulative rewards. Notably, ST-RL-Base-Epoch1, despite exhibiting lower initial rewards during RL training, demonstrates a faster learning rate and subsequently stabilizes at the highest reward level. Combined with Figure 5C, it is evident that models from early SFT stages possess diversity while retaining greater plasticity. This implies that the model has not yet formed overly entrenched biases regarding the data distribution, thereby enabling it to more readily adapt its policy in response to reward signals and engage in broader exploration during the RL phase. These findings indicate that more SFT is not invariably better; rather, a trade-off exists. To achieve optimal outcomes in the subsequent RL training, the selection of a model subjected to a moderate extent of SFT as the initialization point is paramount.

# 6 Generalization to Real-World

To validate the practical effectiveness of our approach, we conduct comprehensive evaluations on real-world GUI tasks to assess the generalization capabilities of agents trained in our simulated environment. This section presents two complementary evaluation strategies: zero-shot evaluation on a static benchmark to test transferability, and continual training experiments to explore scalability with real-world data integration. The detailed specifications of our real-world GUI benchmark and the continual training datasets are provided in Appendix A.5.

## 6.1 Zero-Shot Evaluation on Real-World Static Benchmark

Table 4: Performance on real-world static benchmark with extended action space.

| Action | Total Number | Success Number | Success Rate |
|---|---|---|---|
| CLICK | 971 | 622 | 64.06% |
| COMPLETE | 230 | 221 | 96.09% |
| WAIT | 231 | 99 | 42.86% |
| TYPE | 129 | 119 | 92.25% |
| SCROLL | 8 | 5 | 62.50% |

To demonstrate the practical applicability of our approach, we evaluate the generalization capabilities of agents trained in our simulated environment on real-world GUI tasks. To assess this, we construct a static benchmark by sampling 1,569 instances from several open-source real-world GUI datasets. The test samples cover a broader action space: CLICK, COMPLETE, WAIT, SCROLL, and TYPE. Without any further fine-tuning, we evaluate our agent, which has not encountered WAIT, SCROLL, or TYPE actions during the training process, directly on this benchmark. The results are presented

in the Table 1 below, which show that the agent trained in the simulated environment demonstrates a notable degree of generalization to real-world GUIs. Despite lacking explicit training on WAIT, SCROLL, and TYPE, the agent still achieves non-trivial accuracy on these actions. This suggests that the agent possesses some degree of atomic competence in handling GUI elements beyond its training distribution, highlighting the generalizability of our approach and its practical implications for real-world deployment.

The results shown in Table 4 reveal several key insights: (1) The agent achieves strong performance on COMPLETE (96.09%) and TYPE (92.25%) actions, suggesting effective transfer of fundamental GUI interaction skills. (2) Despite never encountering WAIT, SCROLL, or TYPE actions during training, the agent demonstrates non-trivial accuracy on these unseen actions, indicating emergent generalization capabilities. (3) The moderate performance on CLICK actions (64.06%) reflects the complexity of real-world GUI element recognition, indicating that agents require additional real-world GUI knowledge to improve their understanding.

Table 5: Performance comparison on real-world grounding and interactive benchmarks after continual training on real-world data.

| Model | SS [8] | SS-v2 [43] | FP [19] | MoTIF [4] | Refexp [33] | VWB AG [24] | VWB EG [24] | AW [35] | Aver |
|---|---|---|---|---|---|---|---|---|---|
| Base (paper report) | 84.70 | - | - | - | - | - | - | - | - |
| Base (our report) | 84.01 | 80.34 | 48.25 | 71.93 | 79.46 | 72.81 | 90.07 | 10.34 | 67.15 |
| Continue-Train | 84.91 | 84.43 | 59.50 | 68.30 | 72.13 | 67.96 | 93.70 | 12.06 | 67.87 |
| SFT-Continue-Train | 85.06 | 85.06 | 59.30 | **80.47** | **83.19** | 68.93 | 92.01 | 12.93 | 70.87 |
| ST-RL-Continue-Train | 85.53 | **85.61** | **60.45** | 79.41 | 77.88 | 70.87 | **92.98** | 14.65 | 70.92 |
| MT-RL-Continue-Train | **86.08** | **85.61** | 59.40 | 78.18 | 81.77 | **76.70** | **92.98** | **15.51** | **72.03** |

## 6.2 Evaluation on Real-World Benchmarks with Continual Training

To further investigate the scalability of our approach, we conduct continual training experiments by incorporating real-world data. Specifically, we continue training our agents (SFT, ST-RL, MT-RL) with 24k samples drawn from publicly available real-world GUI datasets. We then evaluate the resulting agents on comprehensive real-world grounding and interactive benchmarks.

As shown in Table 5, our training framework consistently yields agents with strong generalization capabilities across both simulated and real-world GUI tasks. Notably, MT-RL outperforms ST-RL, which in turn outperforms SFT, maintaining the same performance hierarchy observed in our simulated experiments. These results corroborate our main conclusions: (1) MT-RL training yields superior generalization compared to ST-RL training, and (2) ST-RL generalizes better than SFT. The quantitative results show that MT-RL-Continue-Train achieves the highest average performance (72.03%), with particularly strong improvements on ScreenSpot (86.08%), VWB AG (76.70%), and AndroidWorld (15.51%). The consistent performance gains across diverse benchmarks validate the robustness of our multi-turn reinforcement learning approach.

## 7 Conclusion

In this work, we introduce GE-Lab, a novel simulation environment engine designed to advance research in GUI agent navigation. GE-Lab enables flexible and modular construction of screens, icons, and navigation graphs, while granting comprehensive access to environment information for both training and evaluation. Through systematic experimentation, we establish a clear and effective training paradigm: SFT serves as a critical foundation by enabling agents to memorize essential navigation knowledge; ST-RL further enhances generalization to previously unseen scenarios; and MT-RL fosters the development of robust exploration strategies, leading to superior navigation performance and reduced dependence on annotated data. Our extensive benchmarking not only highlights the strengths of reinforcement learning approaches in GUI navigation tasks but also provides actionable insights for building more capable and generalizable GUI agents. We believe that GE-Lab will serve as a valuable resource for the community, facilitating future research and fostering progress towards intelligent and adaptable GUI agents.

## Acknowledgements

This research is supported in part by the National Natural Science Foundation of China (Grant No. U23B2052 and No. 62495092), National Science and Technology Major Project (Grant No. 2021ZD0109803), and National Key Research and Development Program of China (Grant No. 2023ZD0121300).

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

# A   Appendix

## A.1   Experimental Details

This section details the experimental environment, dataset construction methodology, key training parameters, and evaluation setup employed in our research, aiming to ensure the reproducibility and transparency of our findings.

**Compute Resources.**   All experiments are performed on a high-performance computing cluster equipped with 16 NVIDIA A800 GPUs. The specific training duration varies depending on the model stage and task type. A typical training time for a single SFT model is approximately 3 to 4 GPU hours. The training of ST-RL models generally requires about 48 GPU hours, while Multi-Turn Reinforcement Learning MT-RL models typically train for approximately 36 GPU hours.

**Dataset & Benchmark.**   We construct and utilize multiple datasets to support various stages of training and evaluation tasks. Env-Base serves as the core environment for our experiments as shown in Figure 6. This is a graph with a maximum depth of 7. Excluding system-defined edges representing the home and back actions, the branching structure of the environment follows a node distribution of [5, 3, 2, 2, 1, 1, 0] at each respective level. Its graph structure is partitioned into five independent subtrees, which are allocated according to a 2:2:1 ratio for the Supervised Fine-Tuning training set, Reinforcement Learning training set, and test set, respectively. Based on this partitioning, the data are further categorized into path data and edge data. Specifically, each subtree contains 12,439 path data instances and 274 edge data instances. These data are primarily utilized for training the model in graph structure understanding and fundamental navigation capabilities.

In the visual representation, the number of icons corresponds to the clickable areas that trigger page transitions, with the task completed by selecting the "complete" icon. At the root node, there are 5 icons and clickable areas, allowing access to 5 pages. For first-level child nodes, there are 4 icons (3 regular + 1 "Back") and 4 clickable areas, enabling navigation to 4 pages. For other child nodes, which include "Home" and "Back" icons, the number of icons is NodeNum[i] + 2, resulting in NodeNum[i] + 1 clickable areas, allowing access to NodeNum[i] + other pages. On average, each screen contains more than 5 clickable icons.

We additionally generate datasets for specific vision and language understanding tasks: (1) Icon Captioning: This dataset comprises 2,320 instances and is used to train the model to understand icons and generate corresponding textual descriptions. (2) Icon Grounding: This dataset contains 2,320 instances, aimed at training the model to locate relevant icons within an interface based on textual instructions. To evaluate the model's practical performance in dynamic interactive scenarios, we construct a dedicated interactive benchmark task set. This task set is formed by randomly selecting any two nodes within a reserved test environment to serve as the start and end points of a task, respectively. Ultimately, this process generates a total of 2,162 independent interactive test tasks. The test environment is strictly isolated from the training/validation environments to ensure the objectivity of the evaluation. The specific number of tasks, particularly the distribution of tasks according to different path lengths or types, is presented in Table 6.

Table 6: The Number of Interactive Benchmark Tasks

|  | Path@1 | Path@2 | Path@3 | Path@4 | Path@5 | Path@6 | Path@7 | Overall |
|---|---|---|---|---|---|---|---|---|
| Number | 137 | 147 | 222 | 324 | 492 | 456 | 384 | 2162 |

**Details of the Out-of-domain Test Environment.**   To systematically evaluate generalization capabilities, we construct a comprehensive suite of OOD environments that challenge different aspects of visual grounding and navigation. As illustrated in Figure 7, our evaluation framework encompasses five distinct environment configurations: Env-Base (in-domain baseline), Env-Image, Env-Name, Env-Position, and Env-Noise. Following the hierarchical data partitioning strategy outlined in our dataset construction, one subtree is reserved exclusively for OOD evaluation, ensuring zero overlap with training trajectories. The remaining subtrees are allocated for SFT and RL phases, constituting the in-domain training distribution. Beyond this natural distribution shift, we systematically construct four additional OOD variants through controlled perturbations of visual, semantic, and spatial modal-

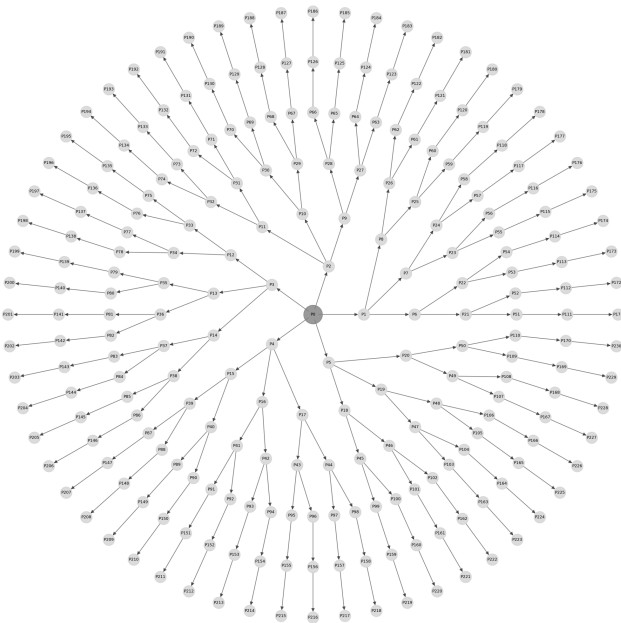

Figure 6: The navigation map of Env-Base.

ities. Each environment preserves the underlying task structure while introducing specific domain gaps that probe distinct aspects of model robustness and transferability.

**Env-Image** introduces visual domain shift by systematically replacing icon appearances while preserving semantic labels and spatial configurations. As depicted in Figure 7, each icon undergoes visual transformation (e.g., the human icon becomes a dog icon) through sampling from a disjoint visual vocabulary, ensuring no visual overlap with the original environment. This perturbation simulates real-world scenarios such as UI redesigns or cross-platform adaptations where functional semantics remain invariant despite visual changes. **Env-Name** targets semantic grounding by altering textual labels while maintaining visual appearance and spatial layout. The transformation involves semantically meaningful substitutions (e.g., "Human" → "Lady") that preserve categorical coherence while introducing lexical variations. This configuration specifically challenges the model's capacity for semantic reasoning and tests the robustness of language-vision alignment mechanisms. **Env-Position** evaluates spatial reasoning capabilities through global layout perturbations. Icon positions undergo unconstrained randomization subject to non-overlap constraints and screen boundary conditions, as illustrated by the repositioned elements in Figure 7. This environment assesses the model's ability to maintain spatial understanding beyond local neighborhood relationships. **Env-Noise** introduces visual complexity through the injection of task-irrelevant distractor elements (Noise1, Noise2). These noise icons are strategically positioned to increase visual clutter while preserving the core navigational structure, thereby testing the model's attention mechanisms and ability to filter relevant visual information.

**Experimental Settings.**    Table 7 and Table 8 present the important hyperparameters used for SFT and RL training. Unless explicitly listed in Table 8, RL hyperparameters match the SFT settings in Table 7.

**Training Time and Sampling Efficiency**    We provide a comprehensive analysis of the training time and sampling efficiency across our three proposed methods. The SFT baseline requires 3-4 GPU hours for training on 60,864 samples, achieving the highest sample efficiency with minimal computational overhead. In contrast, our reinforcement learning approaches demonstrate different computational profiles: ST-RL requires approximately 48 GPU hours for training on 12,439 samples, while MT-RL completes training in 36 GPU hours using 2,162 interactive tasks. The computational trade-offs between methods reflect their distinct learning paradigms. While SFT exhibits superior sample efficiency for in-distribution scenarios, both ST-RL and MT-RL demonstrate enhanced

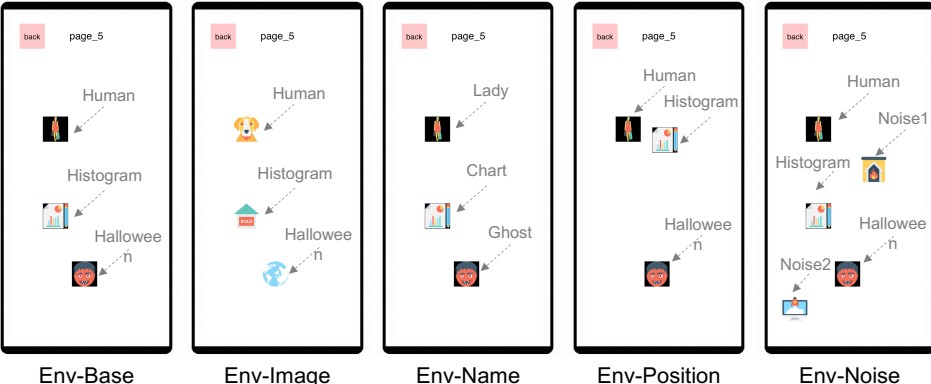

Figure 7: Out-of-Distribution Environment Configurations: Env-Base serves as the in-domain baseline, while Env-Image, Env-Name, Env-Position, and Env-Noise introduce controlled perturbations targeting visual appearance, semantic labels, spatial layout, and visual complexity, respectively. Each environment maintains the same underlying task structure while introducing specific domain shifts to probe different aspects of model robustness.

Table 7: Hyperparameters for SFT.

| config | value |
| --- | --- |
| learning rate | 1e-5 |
| learning rate schedule | cosine decay |
| per device train batch size | 2 |
| gradient accumulation steps | 2 |
| warmup ratio | 0.05 |
| num train epochs | 1 |
| max pixels | 200704 |

Table 8: Hyperparameters for RL.

| config | value |
| --- | --- |
| learning rate | 1e-6 |
| per device train batch size | 8 |
| num train epochs | 5 |
| num generations | 8 |
| temperature | 1.2 |
| top p | 1.0 |
| top k | 8 |

exploration capabilities through increased environment interactions. Specifically, ST-RL achieves 99,512 total interactions compared to MT-RL's 89,939 interactions, with both methods using 8 generations per training round. This increased interaction volume enables more comprehensive state space exploration, leading to improved robustness in out-of-distribution and interactive scenarios.

Regarding the simulation environment, our training framework employs an efficient simulation environment that enables rapid policy iteration without real-time rendering overhead. The environment generates feedback by evaluating the predicted click position against pre-cached meta information, determining state transitions based on spatial overlap with interactive elements. For instance, when the agent predicts a click within the Home icon's bounding box from page_6, the environment transitions to page_0 according to the cached navigation graph. This design provides significant computational advantages: while model inference requires approximately 13 seconds per sample, environment feedback generation incurs only millisecond-level latency due to pre-rendered screen caching. The simulation framework thus enables efficient policy learning without the computational bottlenecks typically associated with real-time GUI rendering, making our approach scalable for large-scale reinforcement learning training.

Table 9: Training Time and Sampling Efficiency Comparison for the Three Methods

| Method | Dataset Size | Training Time | Interaction Count | Feedback |
| --- | --- | --- | --- | --- |
| SFT | 60864 samples | 3–4 GPU hours | 60864 | Manually labeled |
| ST-RL | 12439 samples | 48 GPU hours | 99512 | Manually labeled |
| MT-RL | 2162 tasks | 36 GPU hours | ∼89939 | Environment Feedback |

## A.2 Limitations

While the current version of GE-Lab provides a controlled and accessible environment for studying agent navigation, its interaction logic does not yet fully capture the diversity and complexity of real-world GUI interactions. In practical applications, user interfaces often feature a broader variety of icons, substantial textual information, and various forms of visual noise. These elements, such as additional icons, text, or noise, can be readily incorporated into the current environment, and we plan to continuously enrich the simulation to better approximate real-world scenarios and enhance the generalizability of our findings.

Regarding the action space, we intentionally restrict the agent's available actions in the current version to facilitate focused analysis of navigation strategies. This simplification allows us to systematically evaluate the impact of different training methods on navigation performance. Nevertheless, real-world GUI interactions require a richer set of actions, including gestures such as swiping and long-pressing. Expanding the agent's action space to support more complex interactions is an important direction for future work, enabling more comprehensive assessment of agent capabilities in practical settings.

## A.3 Broader Impact

The advancement of GUI agents hinges critically on their ability to perform robust and generalizable screen navigation, which remains a key bottleneck for real-world deployment. By systematically dissecting the navigation challenge, our work provides a clear roadmap for developing foundational agent capabilities. Specifically, we demonstrate that a staged training paradigm—comprising supervised fine-tuning, single-turn reinforcement learning, and multi-turn reinforcement learning—enables agents to first acquire essential knowledge, then generalize to novel scenarios, and finally develop effective exploration strategies through interactive trial and error.

Our introduction of a flexible simulation environment, GE-Lab, allows for controlled and comprehensive analysis of agent behaviors, circumventing the complexities and proprietary constraints of real-world GUI platforms. This approach not only accelerates research progress by enabling reproducible benchmarking, but also lays the groundwork for agents to safely and efficiently acquire the core competencies required for interacting with complex environments. By abstracting and simplifying the environment, we are able to isolate and address fundamental challenges, ultimately guiding the development of GUI agents toward more general and autonomous intelligence (AGI).

From a societal perspective, the development of more capable and generalizable GUI agents has the potential to significantly enhance productivity and accessibility, automating routine digital tasks and enabling broader access to technology for users with diverse needs. However, as agents become more autonomous and capable of interacting with real-world systems, it is crucial to consider the ethical implications, such as ensuring user privacy, preventing unintended actions, and maintaining transparency in agent decision-making. Our work, by focusing on simulation-based training and evaluation, provides a safer pathway for developing and testing agent capabilities before deployment in real-world applications, thereby helping to mitigate potential risks. We believe that these insights and tools will benefit the broader research community by informing the design of more capable, reliable, and generalizable agents, and by providing a foundation for future work on safe, ethical, and effective agent-environment interaction in both simulated and real-world settings.

## A.4 Reward Function Design

To effectively guide the agent's learning process, we design four complementary reward functions that collectively form the complete reward signal $R$:

$$R(s_t, a_t, s_{t+1}) = r_{type}(a_t, a_t^*) + r_{coord}(a_t, s_t) + r_{intent}(a_t, e_t) + r_{format}(a_t, e_t) \quad (5)$$

where $a_t^*$ represents the reference action, and $e_t$ denotes the generated explanation. We employ equal weights for all four reward components based on empirical analysis of their behavior patterns. Specifically, the Type and Format rewards are consistently optimized early in training, achieving high means with low variance (0.98±0.03 and 0.95±0.05 respectively), effectively serving as syntactic constraints rather than meaningful optimization objectives where heavier weighting would not provide additional gradient information. Meanwhile, the Coordinate and Intent rewards exhibit strong correlation, as evidenced by our analysis of 1000 sampled interactions showing P(R_coord=1 | R_intent=1) $\approx 96.3\%$ and P(R_intent=1 | R_coord=1) $\approx 94.2\%$. This strong dependency between

these reward components reduces the necessity for differentiated weighting schemes, making equal weights a simple yet effective approach that avoids additional hyperparameter tuning.

**Action Type Reward**  The action type reward $r_{type}$ evaluates whether the action type generated by the agent matches the optimal action type. Our action space $\mathcal{A}$ includes two fundamental types: click and complete. Formally, it is defined as:

$$r_{type}(a_t, a_t^*) = \begin{cases} 1, & \text{if type}(a_t) = \text{type}(a_t^*) \\ 0, & \text{if type}(a_t) \neq \text{type}(a_t^*) \end{cases} \tag{6}$$

where $\text{type}(a)$ extracts the type of action $a$. This reward encourages the agent to first understand the action type required in the task context, establishing a foundation for subsequent precise operations.

**Coordinate Accuracy Reward**  The coordinate accuracy reward $r_{coord}$ measures the spatial precision of click operations. For click actions, we determine whether the click coordinates $(x, y)$ fall within the target area based on the bounding box of interactive elements:

$$r_{coord}(a_t, s_t) = \begin{cases} 1, & \text{if type}(a_t) = \text{click and } (x, y) \in \mathcal{P}(s_t) \\ 1, & \text{if type}(a_t) = \text{complete} \\ 0, & \text{otherwise} \end{cases} \tag{7}$$

where $\mathcal{P}(s_t)$ represents the set of points within the correct interactive region in state $s_t$. For complete actions, the default reward is 1 as they do not involve spatial coordinates. This reward mechanism encourages the agent to precisely locate UI elements, enhancing operational reliability.

**Intent Matching Reward**  The intent matching reward $r_{intent}$ evaluates the exact name matching between the explanation $e_t$ generated by the agent and the actual UI element interacted with through action $a_t$:

$$r_{intent}(a_t, e_t) = \begin{cases} \text{ExactMatch}(e_t, \mathcal{I}(a_t, s_t)), & \text{if type}(a_t) = \text{click} \\ \text{Contains}(e_t, \text{``target page''}), & \text{if type}(a_t) = \text{complete} \end{cases} \tag{8}$$

where $\text{ExactMatch}(e_t, \mathcal{I}(a_t, s_t))$ returns 1 if the explanation $e_t$ contains the exact name of the interacted UI element and 0 otherwise. $\mathcal{I}(a_t, s_t)$ represents the name of the specific icon or interactive element that is being clicked when action $a_t$ is executed in state $s_t$. $\text{Contains}(e_t, \text{``target page''})$ returns 1 if the explanation $e_t$ explicitly indicates task completion, and 0 otherwise. For click operations, we verify exact name matching between the element mentioned in the explanation and the actual UI element clicked; for complete operations, we check whether the explanation properly acknowledges the successful completion of the task.

**Format Reward**  The format matching reward $r_{format}$ evaluates whether both the action $a_t$ and the explanation $e_t$ generated by the agent conform to the format of their respective references, $a_t^*$ and $e_t^*$. This reward encourages the agent to produce actions and explanations that are not only correct in content but also consistent in structure and style with the optimal examples. It is formally defined as:

$$r_{format}(a_t, e_t, a_t^*, e_t^*) = \begin{cases} 1, & \text{if FormatMatch}(a_t, a_t^*) \text{ and FormatMatch}(e_t, e_t^*) \\ 0, & \text{otherwise} \end{cases} \tag{9}$$

where $\text{FormatMatch}(a, a^*)$ and $\text{FormatMatch}(e, e^*)$ indicate whether the format of the generated action and explanation match those of the references, respectively (such as structure, key fields, or style). This reward term encourages the agent to generate actions and explanations that are not only semantically correct but also well-formatted and standardized, which benefits subsequent readability, usability, and consistency.

This multi-level reward $R(s_t, a_t, s_{t+1})$ enables the agent to simultaneously learn correct action type selection, precise spatial localization, reasonable semantic understanding, and standardized explanation expression, thereby forming comprehensive navigation capabilities. Experimental results show that after adding the format matching reward, the explanations generated by the agent become more standardized, further improving the navigation success rate and generalization ability in the GE-Lab environment.

## A.5 Real-World Benchmark and Training Data

---

**Evaluation Prompt for Real-World GUI Benchmark**

You are a **Multifaceted Mobile Interface Assistant**. Your responsibilities include:

1. **Navigating** a mobile phone interface to reach a target page based on user instructions, task history, and the current screen state.

2. **Understanding icons** by identifying their name or function based on their location on the screen.

3. **Grounding icons** by locating the coordinates of an icon based on its name or description.

You will receive input that typically includes:

- **User Request:** Specifies the goal (navigation, understanding, or grounding). This might be a complex instruction for navigation or a direct question/command for icon tasks.

- **Task History (Optional, primarily for Navigation):** Records previous steps.

- **Current Screen State:** Represents the current screen, an image (indicated by `<image>`).

**Based on the user request and the current screen state (and history if applicable), you must first determine the type of task requested and then provide the appropriate output.**
**Task Types and Output Formats**
**General GUI Task**

- **Goal:** Reach a target page step-by-step.

- **Typical Input:** Multi-turn instruction, history, and state. screen description and screenshot.

- **Possible Actions:**
    - `click`: Tap a specific element. Provide coordinates (x, y) relative to a (0,0) top-left and (1000,1000) bottom-right system.
    - `type`: Type text into an input field. Provide the string content to be typed. Example: `TYPE("Texas BBQ")`
    - `scroll`: Scroll the screen. Provide the scroll distance. Example: `SCROLL(5)`
    - `wait`: Pause the execution. Provide the duration to wait in seconds. Example: `WAIT(3)`
    - `complete`: Task finished, current screen is the target.

- **Output Format:**

    ```
    Explain:  [Your brief explanation, e.g., 'click xxx
    icon on yyy page.', 'this is the target page.']\tAction:
    [click(start_box=<|box_start|>(x,y)<|box_end|>) or TYPE("Text to type") or
    SCROLL(5) or WAIT(3) or complete]
    ```

**General Instructions**

- Carefully analyze the user request to determine the task (Navigation, Grounding, Understanding).

- Analyze the current screen state (description or image) thoroughly.

- For actions involving coordinates (`click`), use the (0,0) to (1000,1000) system.

- Strictly adhere to the specified output format for the determined task type. Use a tab character (`\t`) as a separator where indicated.

---

**Evaluation Prompt for Real-World Grounding Benchmark**

I want to {goal_info}. Please locate the target element I should interact with. (with point)

---

To comprehensively evaluate the generalization capabilities of our approach, we curate a diverse collection of real-world GUI samples from multiple publicly available datasets. Our real-world evaluation benchmark mentioned in section 6.1 comprises 1,569 carefully sampled instances drawn from four established GUI datasets: AITW [34], AITZ [49], AMEX [5], and Mind2Web [11]. These datasets collectively span diverse application domains, interface designs, and interaction paradigms, providing a comprehensive testbed for cross-domain generalization assessment. The benchmark encompasses five distinct action types: CLICK, COMPLETE, WAIT, SCROLL, and TYPE, representing the fundamental operations in GUI navigation. As depicted in Figure 8, the collected samples exhibit substantial visual and functional diversity, including e-commerce interfaces featuring product listings and shopping carts, system settings with hierarchical menu structures, media applications displaying video content and playback controls, productivity tools with complex layouts, and social platforms incorporating feed-based content and interactive elements. Each sample is annotated with specific instructions and target actions, ranging from simple element selection (e.g., "Click the YouTube icon") to complex multi-step operations (e.g., "Cancel all purchases over $200"), with instructions formulated in natural language that require agents to perform visual grounding, semantic understanding, and precise action execution. The GUI screenshots are sourced from the

original datasets, while the task descriptions and action annotations are generated by GPT-4o-2024-11-20 [29] based on the meta-information from the original datasets, followed by quality assurance conducted by human annotators. The prompt for this task test is shown at A.5 (Evaluation Prompt for Real-World GUI Benchmark). Notably, our simulated training environment only expose agents to CLICK and COMPLETE actions, making WAIT, SCROLL, and TYPE completely unseen during the initial training phase, which enables rigorous evaluation of zero-shot generalization capabilities.

To comprehensively assess GUI understanding and interaction capabilities, we evaluate our approach on eight established real-world benchmarks as discussed in section 6.2. Benchmarks are abbreviated as follows: SS (ScreenSpot) [8], SS-v2 (ScreenSpot-v2) [43], FP (FuncPred) [19], MoTIF [4], Refexp [33], VWB AG (VisualWebBench Agent Grounding) [24], VWB EG (VisualWebBench Element Grounding) [24], and AW (AndroidWorld) [35]. Seven of these benchmarks (SS, SS-v2, FP, MoTIF, Refexp, VWB AG, and VWB EG) are static evaluation datasets that assess visual grounding capabilities by measuring the accuracy of locating target UI elements within screenshots given textual descriptions. In contrast, AndroidWorld (AW) provides an interactive evaluation environment where agents perform complete tasks within Android virtual machines, with success measured by actual task completion rather than intermediate grounding accuracy. This diverse benchmark suite enables thorough evaluation across both fundamental grounding abilities and end-to-end interactive performance. For the continual training experiments described in Section 6.2, we utilize a subset of 24,000 samples from the same source datasets as domain adaptation material, enabling our agents to bridge the gap between simulated and real-world GUI environments while maintaining the learned multi-turn capabilities. For a fair comparison, Qwen2.5-VL-7B-Continue-Train, Qwen2.5-VL-7B-SFT-Continue-Train, Qwen2.5-VL-7B-ST-RL-Continue-Train, Qwen2.5-VL-7B-MT-RL-Continue-Train are trained on the same 24k dataset using 16 GPUs and identical hyperparameters, including a global batch size of 256, a learning rate of 1e-5, a max length of 5120, and 2 epochs. The prompt for the grounding benchmark is A.5 (Evaluation Prompt for Real-World Grounding Benchmark) , and the prompt for the interactive benchmark is A.5 (Evaluation Prompt for Real-World GUI Benchmark).

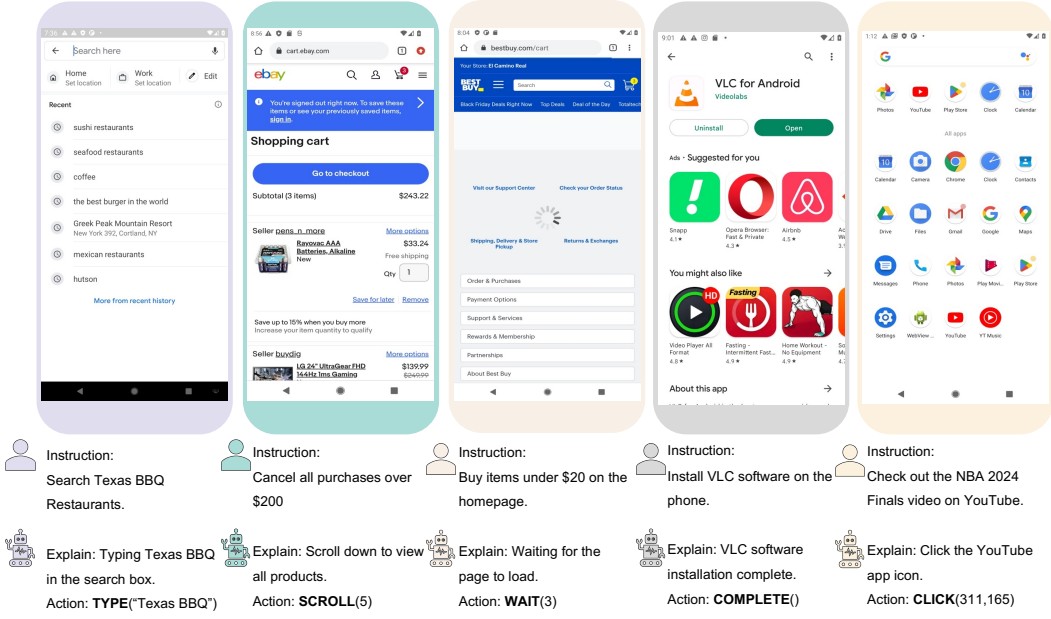

Figure 8: Real-world Data with Expanded Action Space.

## A.6 Training Stability

To evaluate the robustness and reliability of RL methods, we conduct a comprehensive analysis of training stability across multiple random seeds. As demonstrated in Table 10, ST-RL method exhibits remarkably stable training dynamics, with reward values demonstrating a consistent upward trajectory from an initial value of 3.302±0.192 to a final performance of 3.760±0.175. Note that

since ST-RL and MT-RL have different specific iteration numbers, we use It-1 to It-4 to represent different training stages, with the specific correspondence being ST-RL (It_500 → It_2000) and MT-RL (It_200 → It_800). Notably, the standard deviations remain consistently below 0.2 throughout the training process, indicating minimal variance across different experimental runs. Similarly, our MT-RL approach shows substantial and stable improvement, with rewards progressively increasing from 0.394±0.081 to 0.745±0.079. The consistently low variance observed across all experimental configurations provides strong empirical evidence for the stability and reproducibility of our training methodology across different random initializations. Regarding convergence stability, our experimental results reveal that both ST-RL and MT-RL demonstrate smooth and monotonic convergence patterns, which is crucial for practical deployment scenarios. Specifically, the ST-RL approach shows a steady improvement in evaluation scores, progressing from an initial performance of 55.13 to a final score of 63.67, representing a substantial improvement of 8.54 points. Concurrently, the MT-RL method achieves consistent performance gains, with evaluation scores increasing from 63.46 to 64.40. Importantly, we observe that the standard deviations progressively decrease throughout the training process, indicating not only improved performance but also enhanced consistency and reliability. This convergence behavior suggests that our proposed methods successfully avoid common pitfalls in reinforcement learning such as catastrophic forgetting or unstable policy updates, thereby ensuring robust and predictable learning dynamics.

Table 10: Reward and Evaluation Performance Across Iterations for ST-RL and MT-RL (Mean ± Std Over Seeds)

| Model | Type | It-1 | It-2 | It-3 | It-4 |
|-------|------|------|------|------|------|
| ST-RL | Train | $3.302 \pm 0.192$ | $3.693 \pm 0.188$ | $3.739 \pm 0.152$ | $3.760 \pm 0.175$ |
| ST-RL | Eval | $55.133 \pm 1.213$ | $55.358 \pm 1.254$ | $60.077 \pm 1.158$ | $63.663 \pm 1.112$ |
| MT-RL | Train | $0.394 \pm 0.081$ | $0.402 \pm 0.083$ | $0.614 \pm 0.102$ | $0.745 \pm 0.079$ |
| MT-RL | Eval | $63.462 \pm 1.089$ | $63.494 \pm 1.193$ | $64.185 \pm 1.062$ | $64.402 \pm 0.988$ |

**Note:** It-1 to It-4 correspond to: ST-RL (It_500 → It_2000), MT-RL (It_200 → It_800).

## A.7    Reward Hacking in MT-RL

In the exploration of MT-RL, a specific phenomenon of reward hacking is observed, particularly in scenarios where the model possesses a diverse action space and supervision signals are relatively sparse. Figure 5D illustrates the mean output length and reward for two MT-RL models. As indicated by the dashed line, the "MT-RL-w-Hack" model exhibits significant reward hacking in the early stages of training, characterized by a substantial reduction in length alongside a rapid increase in reward. The fundamental cause of this hacking resides in the training mechanism and action space design inherent to MT-RL. Firstly, the primary supervision signals and reward calculations are concentrated on the output of the final turn in multi-turn interactions. Secondly, the model possesses at least two types of actions: "click" and "complete". Typically, task completion must conclude with a "complete" action, a design that leads to a shortcut effect. The model aims to maximize short-term obtainable rewards and tends to select "complete" actions prematurely or excessively. Consequently, it does not genuinely learn how to solve the problem through effective "click" sequences but instead discovers a "loophole" in the reward function. To address this issue, our strategy involves reducing action space diversity to mitigate reward hacking. Specifically, "click" actions are predominantly retained, aiming to guide the model to focus on enhancing the accuracy and effectiveness of these interactions rather than seeking premature task termination. As shown by the solid line in Figure 5D, the "MT-RL-wo-Hack" model demonstrates a stable mean length and a progressive, genuine increase in reward. This indicates that a unified action space enables the model to concentrate on core task actions in each decision-making turn, thereby learning deeper task logic through cumulative multi-turn interactions. These findings offer valuable insights for the future design of more complex MT-RL systems.

## A.8    Case Study

To qualitatively assess the behavioral differences engendered by Supervised Fine-tuning (SFT), Single-Turn Reinforcement Learning (ST-RL), and Multi-Turn Reinforcement Learning (MT-RL), we present 3 illustrative case studies. These studies are conducted within our GE-Lab environment,

where each task has a maximum execution limit of 10 steps. The agent's interactions for each case under SFT, ST-RL, and MT-RL.

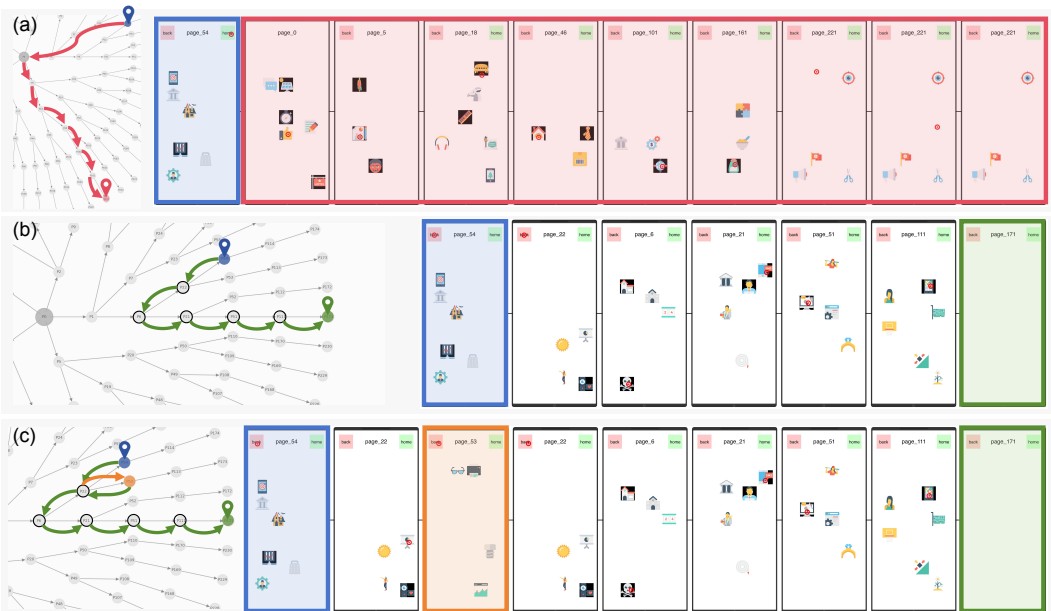

Figure 9: Case Study 1: Demonstrating Basic Navigation and Error Recovery. Task: From page_54 to page_171.

**Case 1**  In the first case, the task involves navigating to a specific target page. The SFT agent (Figure 9(a)) exhibits a characteristic failure mode. It attempts an erroneous direct navigation from the root node to an environment page not contain the target element. Subsequently, it becomes trapped, indicated by futile clicks on a blank area for the final three steps until the episode terminates. This behavior underscores SFT's reliance on memorized trajectories and its brittleness when faced with deviations from seen data, a limitation our training paradigm aims to overcome.

The ST-RL agent (Figure 9(b)) successfully completes the task by identifying and executing the shortest navigation path. This demonstrates the foundational generalization capability imparted by ST-RL, aligning with our contribution that ST-RL enhances generalization to unseen scenarios.

The MT-RL agent (Figure 9(c)) also successfully reaches the target, but notably showcases its enhanced exploratory and recovery capabilities. After an incorrect transition from page_22 to page_53, the agent adeptly recognizes its off-path state, executes a "back" action to return to page_22, and then selects the correct subsequent navigation step. This ability to self-correct and recover from errors highlights MT-RL's promotion of exploration and its capacity to build more robust agents, as states in our contributions.

**Case 2**  The second case presents a similar navigation challenge. The SFT agent (Figure 10(a)) again fails. It initially navigates to a familiar root page before attempting an incorrect direct jump to an unrelated environment page. As in Case 1, it concludes by repeatedly clicking a blank area, exhausting the step limit. This reinforces the observation of SFT's limited generalization.

The ST-RL agent (Figure 10(b)) achieves the goal, albeit imperfectly. While it identifies the shortest path, it makes an initial incorrect click on an invalid area within page_8. Due to GE-Lab's interactive design (where invalid actions result in no state change), the agent remains on page_8 and successfully selects the correct icon on its subsequent attempt, ultimately reaching the target. This shows ST-RL's capacity for in-state error correction.

The MT-RL agent (Figure 10(c)) demonstrates superior performance by navigating to the target via the shortest path with flawless precision in a single attempt per step. This exemplifies the refined decision-making and heightened navigation performance fostered by the MT-RL stage, further corroborating our claims about its benefits.

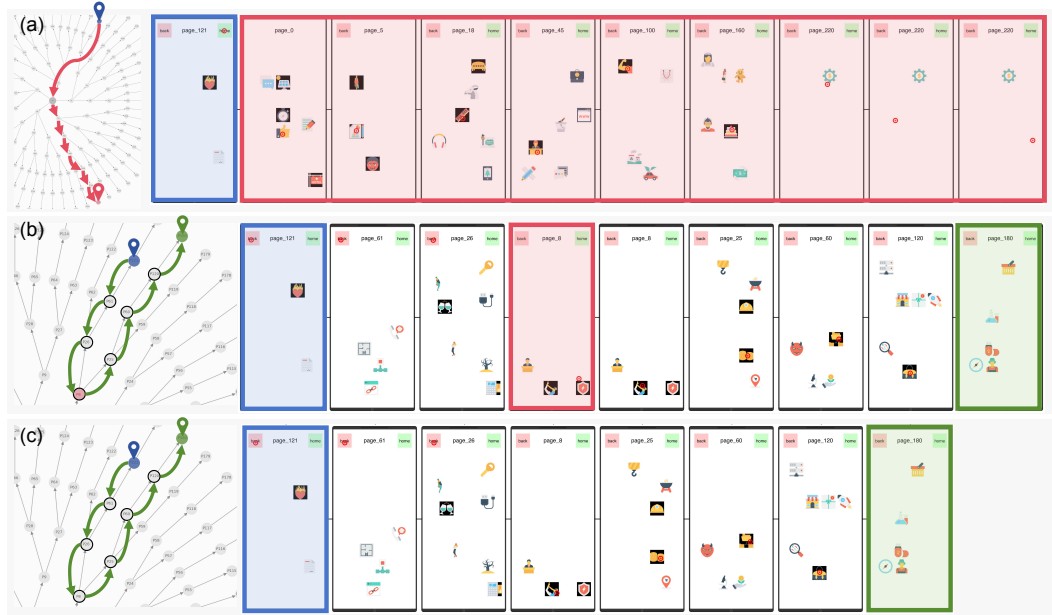

Figure 10: Case Study 2: Navigational Precision and Efficiency. Task: From page_121 to page_180.

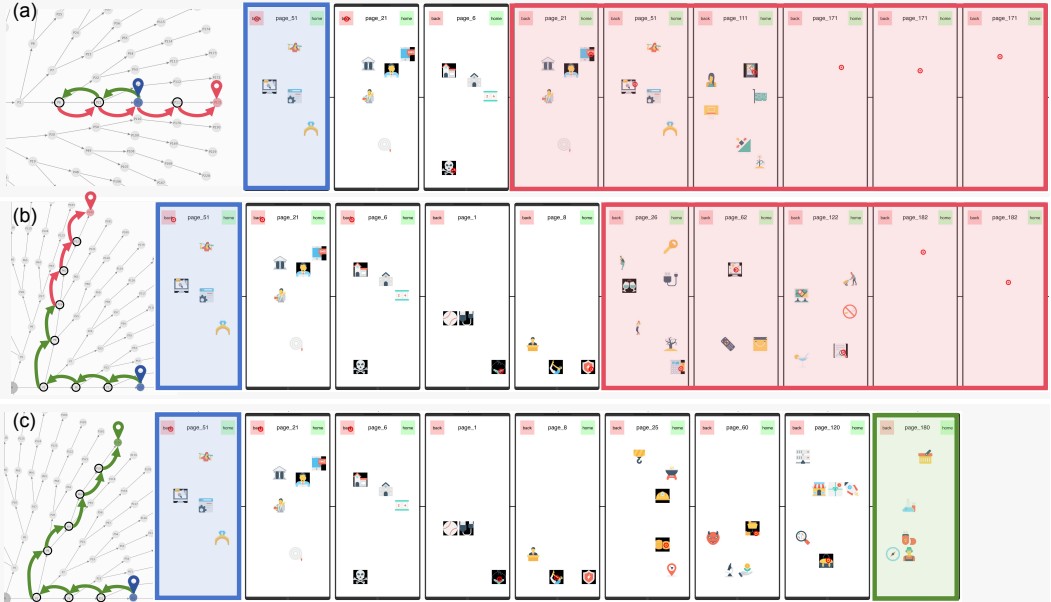

Figure 11: Case Study 3: Complex Navigation and Novel Path Discovery. Task: From page_51 to page_180.

**Case 3** The third case involves a more complex navigation sequence. The SFT agent (Figure 11(a)) initiates the task correctly for the first two steps. However, a single misstep on page_6 (clicking an incorrect icon) leads it down an irretrievable and erroneous path, eventually terminating on a blank page after multiple invalid clicks. This highlights SFT's limitation to recover from navigational errors in longer sequences.

The ST-RL agent (Figure 11(b)) also fails in this more challenging scenario. Despite successfully executing the first five navigation steps, it incorrectly branches off at page_8 and is unable to find the target page before reaching the maximum step limit. This suggests that while ST-RL improves generalization, its exploratory reach in complex, unseen situations may still be limited.

Remarkably, the MT-RL agent (Figure 11(c)) not only successfully reaches the target page but does so via a novel 8-step path that is not present in any of the training data. A critical segment of this path involves a direct and accurate transition from page_51 to page_180. It demonstrates a significant ability to generalize and discover entirely new and efficient solutions. Notably, these discovered solutions are sometimes longer than those typically encountered in training, which strongly supports our contribution that MT-RL promotes exploration. Furthermore, this capability indicates that MT-RL reduces reliance on exhaustive annotated data, as the agent successfully discovers viable paths unseen during the SFT phase.

Collectively, these case studies, facilitated by the flexible GE-Lab environment, provide qualitative evidence for our proposed training paradigm. SFT agents tend to memorize and fail catastrophically upon deviation. ST-RL enhances generalization for moderately unseen scenarios. MT-RL significantly boosts performance, promotes robust exploration and error recovery, and enables agents to discover novel solutions, thereby offering a clear path towards more generalizable GUI agents.

## A.9 Performance Comparison on Tasks of Varying Difficulty

Table 11: Performance Comparison of Methods across Varying Difficulty in Static Benchmarks

|  |  | Path@1 | Path@2 | Path@3 | Path@4 | Path@5 | Path@6 | Path@7 | Overall |
|---|---|---|---|---|---|---|---|---|---|
| **Step** | SFT | 100.00 | 78.005 | 66.216 | 58.457 | 55.928 | 47.556 | 51.302 | 55.454 |
|  | ST-RL | 100.00 | 83.673 | 75.338 | 62.346 | 59.621 | 60.338 | 59.766 | 63.06 |
|  | MT-RL | 100.00 | 79.819 | 72.860 | 64.691 | 59.383 | 60.62 | 60.514 | 63.252 |
| **Task** | SFT | 100.00 | 43.537 | 15.315 | 8.333 | 1.626 | 1.316 | 0.0 | 12.766 |
|  | ST-RL | 100.00 | 57.823 | 24.775 | 13.889 | 3.049 | 2.851 | 0.0 | 16.189 |
|  | MT-RL | 100.00 | 52.381 | 22.072 | 14.506 | 4.268 | 3.07 | 0.26 | 16.003 |

To rigorously evaluate the performance of our proposed methods across tasks of varying complexity, we conduct comprehensive ablation experiments. Table 11 shows the step and task success rates in static benchmark evaluations. We observe a consistent trend across all experimental settings: as the path length increases, the task complexity rises significantly, resulting in performance degradation across all methods. This phenomenon is uniformly present in all testing scenarios, confirming path length as a critical indicator of navigation task difficulty. Reinforcement learning methods (ST-RL and MT-RL) demonstrate markedly superior performance compared to SFT. Notably, at longer path lengths (5-7 steps), ST-RL and MT-RL maintain relatively stable success rates, whereas SFT exhibits a substantial decline. This indicates that reinforcement learning paradigms enhance model generalization capabilities, enabling more robust performance in complex scenarios.

## A.10 System Prompt

---

**System Prompt for Multifaceted Mobile Interface Assistant**

You are a **Multifaceted Mobile Interface Assistant**. Your responsibilities include:

1. **Navigating** a mobile phone interface to reach a target page based on user instructions, task history, and the current screen state.

2. **Understanding icons** by identifying their name or function based on their location on the screen.

3. **Grounding icons** by locating the coordinates of an icon based on its name or description.

You will receive input that typically includes:

- **User Request:** Specifies the goal (navigation, understanding, or grounding). This might be a complex instruction for navigation or a direct question/command for icon tasks.

- **Task History (Optional, primarily for Navigation):** Records previous steps.

- **Current Screen State:** Represents the current screen, an image (indicated by `<image>`).

**Based on the user request and the current screen state (and history if applicable), you must first determine the type of task requested and then provide the appropriate output.**
**— Task Types and Output Formats —**
**1. Task: Navigation**

- **Goal:** Reach a target page step-by-step.

- **Typical Input:** Multi-turn instruction, history, and state. screen description and screenshot.

- **Possible Actions:**

  - `click`: Tap a specific element. Provide coordinates (x, y) relative to a (0,0) top-left and (1000,1000) bottom-right system.
  - `complete`: Task finished, current screen is the target.

- **Output Format:**

  ```
  Explain: [Your brief explanation, e.g., 'click xxx
  icon on yyy page.', 'this is the target page.']\tAction:
  [click(start_box=<|box_start|>(x,y)<|box_end|>) or complete] # Include point
  only for CLICK
  ```

**2. Task: Icon Grounding (Locating an Icon)**

- **Goal:** Identify the coordinates of a requested icon.

- **Typical Input:** User request like "Click on [icon name/description] in the image.", screen image (`<image>`).

- **Action:** Implicitly `click` (meaning "identify location").

- **Output Format:** The explanation is often implicit in the grounding request itself.

  ```
  Action: click(start_box=<|box_start|>(x,y)<|box_end|>)
  ```

**3. Task: Icon Understanding (Identifying an Icon)**

- **Goal:** Provide the name or function of an icon at given coordinates.

- **Typical Input:** User request like "What is the icon at point (x, y) in the image?", screen image (`<image>`).

- **Action:** Provide textual information.

- **Output Format:** Just the direct answer as text.

  ```
  [Icon Name or Description]
  ```

**— General Instructions —**

- Carefully analyze the user request to determine the task (Navigation, Grounding, Understanding).

- Analyze the current screen state (description or image) thoroughly.

- For actions involving coordinates (`click`), use the (0,0) to (1000,1000) system.

- Strictly adhere to the specified output format for the determined task type. Use a tab character (`\t`) as a separator where indicated.

---

