# OpenReview forum: "GUI Exploration Lab: Enhancing Screen Navigation in Agents via Multi-Turn Reinforcement Learning"
_NeurIPS.cc/2025/Conference — NeurIPS 2025 poster_

### Official Review · Reviewer_JzUR · 2025-06-24

**Clarity:** 2
**Significance:** 2
**Originality:** 4
**Rating:** 4
**Confidence:** 2

**Summary:**

The paper considers the pages in GUI as graph and apply single-turn and multiple-turn reinforcement learning to learn and explore the GUI environment.
The proposed approach enables the agent to learn structured navigation behaviors. The agent is also to emulate user actions across different task configurations.

**Questions:**

- How exactly is the OOD test environment constructed? What makes it OOD compared to the ID training set? Please clarify with specific differences, e.g., layout, content, structure.
- For complex tasks, how long can the history string be? Is there a limit imposed, and if so, how is truncation handled?
- In Equation (2), the reward function is composed of four parts, but none are mathematically defined. Please provide a formal definition of each component and explain how they are computed. And what is the weight factor, $\lambda$, for each component?
- RL is known to exhibit high variance. The readers would appreciate it if they can see the stability of the use of RL in this case, which can be demonstrated in Fig. 5 by adding shaded variance regions.

**Ethical Concerns:**

["NO or VERY MINOR ethics concerns only"]

**Final Justification:**

My concerns regarding the strategy for generating the OOD settings have been addressed, and I now recognize the impact of this submission. I am updating my overall rating from 2 to 4, and the originality score from 2 to 4 as well. However, the current manuscript still suffers from clarity issues, including typos and dense methodological explanations. If the chairs are considering accepting this submission, I would recommend that a thorough revision be requested to improve clarity and readability.

**Limitations:**

Yes

**Paper Formatting Concerns:**

I do have some concerns about the citations and references in this paper.

- The reference by Dell’Aversana in line 102 is not peer-reviewed and does not explain reinforcement learning. And neither the OpenAI gym by Brockman et. al. in the same line does. They are not suitable for the claim about RL.
An example of the more appropriate references would be: *Sutton, R. S., & Barto, A. G. (2018). Reinforcement learning: An introduction. MIT press*.

- line21: author information is missing in the in-line citation format. You should instead use the title of the paper as the author, such as "Claude 3".

- Some of the in-text citations have a missing year (line 88), and some have a non-alphabetical symbol in the author name (line 62).

- In the references, the capitalization of the name in title should be noticed, such as line 375 "GUI", line 411 - "GPT-4o," and line 456 - "OSWorld". Since many multimodal agent papers include capitalized terms or branded names in the titles, it would be better to keep the capitalization consistent across all references.

- The author uses both `arXiv e-prints` (line 390) and `arXiv preprint arXiv:yymm.xxxxx` (line 393). Consistency should be maintained.

- For conference proceedings, it should be, for example, "**In** Advances in Neural Information Processing Systems" rather than listing the conference as if it were a journal. (line 367, 420, 423, and 457)

**Quality:**

2

**Strengths And Weaknesses:**

Strength:

- The author provides a simulation environment engine that can generate a GUI environment and proposes to try RL methods to explore it.

Weaknesses:
- The author adopted a randomized placement of the icons. However, this setting may not reflect realistic UI design principles. In real-life practical situations, GUIs tend to follow structured and intuitive layouts to enhance usability. Shouldn't it be more suited to learn icons with placement that is similar to modern-day UI design to complete its task to be useful?
Training agents on fully randomized interfaces, even for the sake of OOD robustness, may result in behavior that is less transferable to real-world systems. Maybe it would be more informative to consider a setting that maintains a plausible UI design and structure.
- There is little description about the engine that generates the GUI learning environment for LLM agents. How the engine works is not clear; for example, what is the generated GUI environment like? how much information and available actions are on the visible page? how many other pages can one page reach on average?

**nit**
- line35: "mobile apps"
- Fig.1(c) Is it true that all pages (the node) have only a degree of 1, i.e., they only have a distinct neighbor? The illustration is not very clear.
- For Fig. 2, I would suggest a top-down, left-to-right ordering of the A, B, and C components. Current ordering makes the visual confusing.
- line 108: typo "extende"

---

> ### Author Rebuttal · Authors · 2025-07-31
>
> We would like to thank the reviewer‘s constructive feedback. **We appreciate the recognition of our simulation engine as a valuable tool for generating and exploring GUI environments.** We also acknowledge the reviewer’s concern about the randomized icon placement and its transferability to real-world systems. As discussed in the following section, we attempt to address this issue by considering more structured UI designs in our experiments to maintain a balance between OOD robustness and practical relevance. Additionally, we have provided further clarification on the workings of the engine and its generated environments. The feedback has greatly helped us in refining these aspects.
>
> Below are our detailed responses to Reviewer JzUR.
>
> 1. **Weakness 1:** Shouldn't it be more suited to learn icons with placement that is similar to modern-day UI design to complete its task to be useful? Whether randomized icon placement affects generalization in real-world environments.
> - Response:
>   * The decision to randomly sample and place icons in the simulated environment is intentional and driven by the goal of promoting generalization. If the icons were not randomized, the agent might memorize which icons to click for specific tasks, making it difficult to evaluate the agent's memorization for the navigation map. Similarly, if the positions were not randomized, the model might overfit to specific data points, memorizing fixed positions and clicking on. These would hamper the evaluation of the agent's grounding ability. By contrast, **if the agent is still able to correctly complete tasks in the randomized setting, this would strongly demonstrate that it possesses a solid screen navigation ability.**
>   * Importantly, we have also validated the agent's performance in real-world GUI environments, where layouts tend to follow more structured, modern UI design patterns. **The results showed that the agent can successfully perform screen navigation in well-structured real-world settings.** Specifically, we continue training the agents of three methods with 24k samples drawn from publicly available datasets, including AITW [1], AITZ [2], AMEX [3], and Mind2Web [4]. We then evaluated the resulting agents on both real-world grounding and interactive benchmarks. As shown in Table 1, MT-RL achieves higher accuracy than ST-RL, which in turn outperforms SFT. This suggests that the agents trained in the simulated environment are able to generalize effectively to real-world benchmarks, encompassing both grounding and interactive tasks.
>
> **Table 1:** Performance of the agent on real-world grounding benchmark and interactive benchmark.
>
> |Model|ScreenSpot[5]|ScreenSpot&#8209;v2&nbsp;[6]|FuncPred[7]|MoTIF[8]|Refexp[9]|VWBAG[10]|VWBEG[10]|AndroidWorld[11]|Average|
> |-|-|-|-|-|-|-|-|-|-|
> | Qwen2.5-VL-7B-Base(paper report) |84.70|-|-|-|-|-|-|-|-|
> |Qwen2.5-VL-7B-Base(our report)|84.01|80.34|48.25|71.93|79.46|72.81|90.07|10.34|67.15|
> |Qwen2.5-VL-7B-Continue-Train|84.91|84.43|59.50|68.30|72.13|67.96|93.70|12.06|67.87|
> |Qwen2.5-VL-7B-SFT-Continue-Train|85.06|85.06|59.30|**80.47**|**83.19**|68.93|92.01|12.93|70.87|
> |Qwen2.5-VL-7B-ST-RL-Continue-Train|85.53|**85.61**|**60.45**|79.41|77.88|70.87|**92.98**|14.65|70.92|
> |Qwen2.5-VL-7B-MT-RL-Continue-Train|**86.08**|**85.61**|59.40|78.18|81.77|**76.70**|**92.98**|**15.51**|**72.03**|
>
> 2. **Weakness 2:** How the engine works is not clear; for example, what is the generated GUI environment like? how much information and available actions are on the visible page? how many other pages can one page reach on average? Details of the GE-Lab engine and the simulated environment.
> - Response:
>   * GE-Lab is a novel simulation environment engine. Specifically, GE-Lab first **generates a graph** of the environment based on user-defined parameters, which determines the navigation relationships between different screens. Then, GE-Lab **renders a screen** for each node of the graph, with each screen displaying n icons (where n is the degree of the node). Clicking an icon triggers a page transition. The icons used in GE-Lab are randomly sampled from an icon library, enhancing the diversity of the simulation environment.
>   * The simulated environment used for our main experiments is Env-Base, as described in Section 4.1. **This is a graph with a maximum depth of 7**. Excluding system-defined edges representing the home and back actions, the branching structure of the environment follows a node distribution of [5, 3, 2, 2, 1, 1, 0] at each respective level. Figure 6 visualizes this graph structure. The graph is partitioned into five independent subtrees, which are allocated to the SFT training set, RL training set, and test set in a 2:2:1 ratio, respectively.
>   * In the visual representation, the number of icons corresponds to the clickable areas that trigger page transitions, with the task completed by selecting the "complete" icon. At the root node, there are 5 icons and clickable areas, allowing access to 5 pages. For first-level child nodes, there are 4 icons (3 regular + 1 "Back") and 4 clickable areas, enabling navigation to 4 pages. For other child nodes, which include "Home" and "Back" icons, **the number of icons is NodeNum[i] + 2, resulting in NodeNum[i] + 1 clickable areas**, allowing access to NodeNum[i] + other pages. On average, each screen contains more than 5 clickable icons.
>
> 3. **Question 1:** How exactly is the out-of-domain (OOD) test environment constructed? What makes it OOD compared to the  in-domain (ID) training set?  The construction details of the OOD environment and its differences from the in-domain environment.
> - Response:
>   * As described in Dataset & Benchmark (A.1, lines 483-487), **one of the five subtrees under the root node is excluded from both the training and test sets, representing an OOD environment.** The path data from the other two subtrees are used for SFT training, while the remaining two subtrees contribute path data for RL training, which are considered in-domain environments. The OOD samples involve paths that were not encountered during the agent's training process.
>   * In addition, we have constructed other **more challenging OOD environments**. As described in Section 4.1, we generated Env-Image, Env-Name, and Env-Position by augmenting the icon’s image, name, and position, respectively. Furthermore, we introduced noisy icons to create the Env-Noise environment.
>   * In terms of the differences between OOD and ID environments, the icons on both types of screens are randomly generated and positioned at different nodes in the graph. As a result, the screen layouts, content, and structure of the two environments are distinct.
>
> 4. **Question 2:** For complex tasks, how long can the history string be? Is there a limit imposed, and if so, how is truncation handled? Context string length constraints and truncation strategies.
> - Response:
>   * In both multi-turn RL online training and actual inference, **we set a maximum number of interaction rounds** (e.g., 12). According to our agent protocol (Section 3.2.2), the context token length, composed of the text from a single dialogue round and the current observation image, is less than 1k tokens, which is within the model's context length. Therefore, **no additional truncation is required.**
>
> 5. **Question 3:** Please provide a formal definition of each component and explain how reward are computed. And what is the weight factor for each component?
> - Response:
>   * The reward function formula is **provided in Appendix A.4**. Thank you for your suggestion; we will add a reference to the section containing the details of the reward function in Section 3.3.1.
>
> 6. **Question 4:** RL is known to exhibit high variance. The readers would appreciate it if they can see the stability of the use of RL in this case, which can be demonstrated in Fig. 5 by adding shaded variance regions. The variance in RL training metrics requires visualization to demonstrate training stability.
> - Response:
>   * Thank you for the reminder. **We will include the variance information in Figure 5** in the final version.
>
> 7. **Nit 2:** In Figure 1C, is it true that all pages (the node) have only a degree of 1, i.e., they only have a distinct neighbor? The illustration is not very clear. Illustration of node diagrams in Figure 1(C,D,E).
> - Response:
>   * To clarify, in Figure 1C, **only nodes with degree 1 represent training data that contains knowledge about two connected nodes.** This does not imply that every node in the actual state transition graph has a degree of 1.
>   * Figure 1C visualizes all training samples as nodes with degree 1, indicating that the **SFT training for a single sample only memorizes knowledge of a one-step transition**. As shown in Figure 2A, SFT samples follow the data protocol of POMDP, and some of them contain not only one-step transition knowledge (corresponding to degree 1), but also historical trajectories in the history.
>
> 8. **Nit 3:** For Figure 2, I would suggest a top-down, left-to-right ordering of the A, B, and C components. Current ordering makes the visual confusing.
> - Response:
>   * Thank you for the suggestion. **We will adjust the numbering sequence** in Figure 2 accordingly.
>
> 9. **Nit 1 & Nit 4:** Typo.
> - Response:
>   * Thank you for your thorough review. We will make the necessary revisions and perform another round of checks.
>
> 10. **Paper Formatting Concerns:** Reference formatting and authority verification.
> - Response:
>   * **We will thoroughly review all references, confirm their publication status, and ensure the quality of the journals in which they appear.** This includes addressing issues related to "Improper citation of foundational RL concepts" and "Formatting errors in in-text citations." Due to space limitations, we will present the final references in the revised version.
>
> **Reference**
>
>  Due to space limitations, please refer to our response to Reviewer DUjd for details regarding the references.

---

> > ### Comment · Reviewer_JzUR · 2025-08-04
> >
> > Thank you to the authors for the detailed and thoughtful rebuttal. I really appreciate the clarifications provided.
> >
> > That said, I still have a few points that need further clarification:
> > 1. I understand that the authors can't upload any plots, but I would still like to see some textual descriptions on:
> >     - How stable was the training across seeds?
> >     - Were there major fluctuations or signs of convergence instability, especially in MT-RL?
> >
> > 2. While I understand that *Env-Image, Env-Name, and Env-Position* refer to variations in icon appearance, label, and layout, the details of these modifications are still unclear: Could the authors explain concretely how these variations are implemented? For example: Are images randomly swapped from a visual vocabulary? Are names altered semantically or syntactically? Are positions perturbed in a constrained or unconstrained manner? And I believe a graphical example would be very helpful here, so that the reader can read Table 3 better.
> >
> > 3. While randomization may help prevent overfitting, I wonder if aligning placements more closely with human UI design principles (e.g., thumb-reachable zones, functional grouping, padding) might result in agents that better generalize to structured real-world layouts. Have the authors explored or considered this trade-off? Would these issues affect the agent's performance or learning stability?
> >
> > 4. The reward function is treated with equal weight. Is there a specific rationale for keeping all weights equal?
> >
> > I've adjusted my score, but again, my concerns are not fully addressed, and I look forward to further clarifications.

---

> > > ### Author Response · Authors · 2025-08-04
> > > **Response to Reviewer JzUR**
> > >
> > > We would like to thank the reviewer JzUR for the insightful comments across multiple aspects of our work.
> > >
> > > **Question1: I understand ...**
> > > * Response
> > >   * **Training stability**: As shown in **Table 1**, ST-RL rewards increase steadily from 3.302 ± 0.192 to 3.778 ± 0.178 with stds < 0.2. MT-RL rewards grow from 0.394 ± 0.081 to 0.752 ± 0.081. This consistent upward trend and low variance indicate stable training dynamics across seeds.
> > >   * **Convergence stability**: Both methods show smooth, monotonic improvements. ST-RL's eval score rises from 55.13 to 63.73, MT-RL from 63.46 to 64.39, with decreasing stds. We will include variance curves in the final revision.
> > >
> > > **Table 1 Reward and Evaluation Performance Across Itations for ST-RL and MT-RL (Mean ± Std Over Seeds)**
> > > |Model|Type|It-1|It-2|It-3|It-4|It-5|
> > > |-|-|-|-|-|-|-|
> > > |ST-RL|Train|3.302±0.192|3.693±0.188|3.739±0.152|3.760±0.175|3.778±0.178|
> > > |ST-RL|Eval|55.133±1.213|55.358±1.254|60.077±1.158|63.663±1.112|63.735±1.015|
> > > |MT-RL|Train|0.394±0.081|0.402±0.083|0.614±0.102|0.745±0.079|0.752±0.081|
> > > |MT-RL|Eval|63.462±1.089|63.494±1.193|64.185±1.062|64.402±0.988|64.386±0.885|
> > >
> > > **Note**: It-1 to It-5 correspond to: ST-RL (It_500 → It_2500), MT-RL (It_200 → It_1000).
> > >
> > > **Question2: While I ...**
> > > * Response
> > >   * **Env-Image**  changes the visual appearance of icons while keeping their names and positions fixed. Each icon is replaced with another from a fixed vocabulary, sampled such that it does not appear elsewhere in the environment. This simulates cases like UI updates where icon appearance changes but function remains the same.
> > >   * **Env-Name** alters the semantic name of each icon while keeping its visual appearance and position fixed. The name changes are meaningful (e.g., replacing “Hotel” with “Restaurant”), reflecting scenarios where visually similar icons are repurposed across apps but assigned different textual labels. This setting challenges the agent’s reliance on semantic cues for grounding.
> > >   * **Env-Position** randomly perturbs icon locations while keeping appearance and name fixed. Perturbations are unconstrained globally, with only two rules: (1) no overlap (including with system icons), and (2) icons must remain within screen bounds. This tests robustness to layout variation beyond local shifts.
> > >   * To help interpret Table 3 in paper, we will include a graphical illustration in the revision showing how a screen appears under each of the four environment settings.
> > >
> > > **Question3: While randomization ...**
> > > * Response
> > >   * To test whether human-like UI layouts improve generalization, we compared models trained with fixed (grid-aligned) icon layouts to those with random positions.
> > >   * As shown in Table 2, the (fix) setting uses neatly arranged icons resembling real UI designs, while (random) retains our default randomized layout. Across all methods, performance on real-world benchmarks is very similar, with (random) often slightly better. This suggests that layout regularity in simulation does not significantly enhance generalization.
> > >   * We also clarify that GE-Lab Grounding (Table 2) corresponds to the Icon Grounding task in Appendix A.1. Prompts were augmented to avoid memorization. All models achieve ~99–100% accuracy, confirming that grounding on seen icons is robust, and navigation policy learning is the key challenge.
> > >   * As shown in Figure 3, our simulated UI only includes page names, system icons, and task icons—**without** thumb-reachable zones, grouping, or padding. Hence, we cannot make claims about their effect on generalization.
> > >   * We agree reducing the domain gap is valuable, but constructing realistic yet controllable UI simulations at scale remains a **challenging open problem**. Our focus is to study RL-based screen navigation in a simplified environment that **minimizes confounding factors.**
> > >
> > > **Table 2:** Benchmark Results for Fix vs. Random Icon Positions
> > > |Model(Layout)|ScreenSpot-v2[6]|AndroidWorld[11]|GE-Lab Grounding|
> > > |-|-|-|-|
> > > |SFT(fix)|85.10|12.93|**99.56**|
> > > |SFT(random)|85.06|12.93|99.53|
> > > |ST-RL(fix)|85.56|13.79|99.21|
> > > |ST-RL(random)|**85.61**|14.65|99.30|
> > > |MT-RL(fix)|85.50|**15.51**|99.28|
> > > |MT-RL(random)|**85.61**|**15.51**|**99.47**|
> > >
> > > **Question4: The reward ...**
> > > * Response
> > >   * Our use of equal weights is based on empirical analysis of reward behavior:
> > >     * Type and Format rewards are consistently optimized early, with high means and low variance (0.98 ± 0.03, 0.95 ± 0.05), acting more as syntactic constraints than useful meaningful objectives. Heavier weighting would not provide additional gradient.
> > >     * Coordinate and Intent rewards are strongly correlated. From 1000 sampled interactions, we observe:
> > > |Total: 1000|R_Intent=1|R_Intent=0|
> > > |-|-|-|
> > > |R_Coord=1|385|22|
> > > |R_Coord=0|15|578|
> > >
> > >       P(r_coord = 1 | r_intent = 1) ≈ 96.3%, and P(r_intent = 1 | r_coord = 1) ≈ 94.2%. **Their strong dependency reduces the need for differentiated weighting**. Equal weights thus offer a simple and effective structure without added tuning.

---

> > > > ### Comment · Reviewer_JzUR · 2025-08-05
> > > >
> > > > Thank you to the authors for their discussion. I have adjusted my confidence and rating accordingly. Please ensure that the points raised during our discussion are included in the revision.

---

> > > > > ### Author Response · Authors · 2025-08-06
> > > > > **Response to Reviewer JzUR**
> > > > >
> > > > > We sincerely thank the reviewer JzUR for their thorough and constructive feedback on our paper and rebuttal. We will include the points raised during the discussion in the final version.

---

### Official Review · Reviewer_VsGX · 2025-06-30

**Clarity:** 3
**Significance:** 2
**Originality:** 2
**Rating:** 4
**Confidence:** 5

**Summary:**

The paper introduces GUI Exploration Lab (GE-Lab), a synthetic environment for GUI agent navigation. It constructs a graph where each page has icons that, when clicked, lead to other nodes (pages). The tasks are straightforward: navigate from page X to page Y. The authors test three training methods—supervised fine-tuning (SFT) to memorize paths, single-turn reinforcement learning (ST-RL) for generalization, and multi-turn reinforcement learning (MT-RL) for exploration. Experiments show MT-RL performs best in interactive settings. The paper claims these findings guide GUI agent development, but the reviewer finds the work lacks significance due to its oversimplified approach.

**Questions:**

**Does training in GE-Lab actually improve agent performance in real GUI environments?** If so, provide data or reasoning to support this. My main concern is that the conclusions drawn in this paper can also be drawn from prior work due to the lack of uniqueness from the GUI side. I will only be convinced and give higher marks if this environment is proven useful for enhancing model performance on realistic tasks or is sufficiently realistic for evaluation.

**Ethical Concerns:**

["NO or VERY MINOR ethics concerns only"]

**Final Justification:**

The additional experiments partially address my concerns about transfer learning and practical impact. The results in Tables 1 and 2 do demonstrate some effectiveness of GE-Lab for real-world GUI tasks, and I adjust my score accordingly.

**Limitations:**

Yes

**Paper Formatting Concerns:**

No major formatting issues were noted in the critique, and the paper is assumed to comply with NeurIPS 2025 formatting instructions.

**Quality:**

2

**Strengths And Weaknesses:**

**Strengths:**
- **Originality**: The introduction of GE-Lab is a novel contribution, providing a controlled environment for GUI navigation research. The systematic comparison of SFT, ST-RL, and MT-RL training paradigms offers a structured approach to agent development, which is a fresh perspective in the field.
- **Quality**: The experimental design is technically sound, with detailed results (e.g., Table 1 shows MT-RL achieving Pass@1 of 17.47 and Pass@5 of 25.16) and a clear methodology using Qwen2.5-VL-7B in a tree-structured environment (depth 7). The use of both static and interactive benchmarks adds rigor.
- **Clarity**: The paper appears well-structured, with figures (e.g., Figure 1, Figure 2) and tables (e.g., Table 1, Table 2) likely aiding understanding. No clarity issues were raised in the critique, suggesting effective communication of methods and results.
- **Significance**: The work addresses a relevant challenge in GUI agent navigation, a growing area with LVLMs. The benchmarking and training insights could inform future research, though practical impact is limited.

**Weaknesses:**
- **Quality**: The paper oversimplifies GUI interactions by reducing navigation to clicking icons, ignoring complex actions like scrolling, text input, right-clicking, or double-clicking (Appendix A.2 acknowledges plans to expand action space, but current work is limited). This simplification undermines the robustness of the findings for real-world applications.
- **Significance**: The lack of evidence for transfer learning from GE-Lab to real GUI environments significantly reduces practical impact. Performance drops in OOD settings (e.g., Table 3, Env-Image: 34.17 for MT-RL vs. 63.25 in Env-Base) suggest limited generalization, aligning with concerns that synthetic environments may not translate to real-world improvements.
- **Originality**: While GE-Lab is novel, the concept of synthetic navigation environments echoes prior work (e.g., text-based games), and the conclusions (e.g., RL enhances exploration) are not entirely new, reducing the paper’s distinctiveness.
- **Clarity**: Although generally clear, the paper’s failure to address how its simplified model applies to complex GUI tasks may confuse readers expecting real-world relevance.

Ammanabrolu, P., & Riedl, M. (2019). Playing Text-Adventure Games with Graph-Based Deep Reinforcement Learning. In Proceedings of the 2019 Conference of the North American Chapter of the Association for Computational Linguistics: Human Language Technologies, Volume 1 (Long and Short Papers), pages 3557–3565, Minneapolis, Minnesota. Association for Computational Linguistics.

Osborne, P., Nõmm, H., & Freitas, A. (2022). A Survey of Text Games for Reinforcement Learning Informed by Natural Language. Transactions of the Association for Computational Linguistics, 10, 759-776. https://doi.org/10.1162/tacl_a_00495

Gur, I. (2018). Learning to Navigate the Web. arXiv preprint arXiv:1812.09195.

Li, Y., & Riva, O. (2021). Glider: A Reinforcement Learning Approach to Extract UI Scripts from Websites. In Proceedings of the 44th International ACM SIGIR Conference on Research and Development in Information Retrieval, 1420–1430.

---

> ### Author Rebuttal · Authors · 2025-07-31
>
> We would like to thank the reviewer for the thoughtful feedback. **We appreciate the recognition of GE-Lab as a novel contribution to the field, and we also value the reviewer's acknowledgment of the technical soundness and clarity of our experimental design.** Regarding the practical impact, we have now elaborated on how the insights from our work can inform future research and practical applications in the following section. The feedback has been invaluable in helping us refine and strengthen our manuscript.
> Below are our detailed responses to the Reviewer VsGX.
>
> 1. **Weakness 1 Quality:** This simplification undermines the robustness of the findings for real-world applications.
> - Response:
>   * To address the concern about whether the simplification of GUI interactions might limit the robustness of our finding, **we expanded the action space to assess the agent’s ability to perform a broader range of actions.** Specifically, we sampled 1,569 instances from an open-source real-world GUI dataset (including AITW [1], AITZ [2], AMEX [3], and Mind2Web [4]) and manually constructed a static benchmark. The test samples in this benchmark cover the following 5 actions: CLICK, COMPLETE, WAIT, SCROLL, and TYPE, the last three of which were not included in the previous action space. We then evaluated the agent without any fine-tuning on this extended benchmark. As shown in Table 1, the agent achieved non-trivial accuracy on WAIT, SCROLL, and TYPE actions, despite not encountering these three new actions during the training process. This indicates that the agent has acquired transferable atomic skills for interacting with GUI elements beyond its original action space, supporting the potential applicability of our approach to more complex real-world scenarios.
>
> **Table 1:** Performance of the agent on real-world static benchmark with extended action space.
>
> | Action   | Total Number | Success Number | Success Rate |
> | ---------- | -------------- | ---------------- | -------------- |
> | CLICK    | 971          | 622            | 64.06%       |
> | COMPLETE | 230          | 221            | 96.09%       |
> | WAIT     | 231          | 99             | 42.86%       |
> | TYPE     | 129          | 119            | 92.25%       |
> | SCROLL   | 8            | 5              | 62.50%       |
>
> 2. **Weakness 2 Significance & Question 1:** Whether the lack of evidence for transfer learning from GE-Lab to real GUI environments significantly reduces practical impact. & **Does training in GE-Lab actually improve agent performance in real GUI environments?**
> - Response:
>   * We further trained the SFT agent, ST-RL agent, and MT-RL agent on real-world datasets (including AITW [1], AITZ [2], AMEX [3], and Mind2Web [4]), and the accuracy results are shown in the Table 2. It is evident that **the agent's performance on the real-world dataset exceeds the baseline**, demonstrating that GE-Lab can improve agent performance in real-world GUI environments. We believe this can address the reviewer’s concern about the practical impact and the major question that the reviewer raised.
>   * As shown in Table 2, all agents trained in GE-Lab and then fine-tuned on real-world data outperform both the base model (Qwen2.5-VL-7B-Base) and the model that continued training directly on real-world data without GE-Lab pretraining (Qwen2.5-VL-7B-Continue-Train). Notably, **the MT-RL agent achieves the best performance across most benchmarks**, with the highest accuracy on ScreenSpot (86.08%), and Refexp (81.77%), as well as strong results on interactive benchmarks like AndroidWorld (15.51%). The average performance of the MT-RL agent is 72.03%, surpassing the other models in overall accuracy, reflecting its robustness.
>   This demonstrates that reinforcement learning in GE-Lab provides generalizable screen navigation capabilities that effectively transfer to complex real-world tasks. The consistent performance gains from SFT to ST-RL to MT-RL, as seen in both the individual benchmarks and the overall average, mirror our simulation findings. These results offer strong empirical support that the skills acquired in GE-Lab are transferable and beneficial when applied to real-world GUI environments, both in grounding and interaction settings.
>   * It is worth noting that the results above address both the concern regarding the significance of this work and the major question raised by the reviewer: "Does training in GE-Lab actually improve agent performance in real GUI environments?"
>
> **Table 2:** Performance of the agent on real-world grounding benchmark and interactive benchmark.
>
> | Model                              | ScreenSpot [5] | ScreenSpot-v2 [6] | FuncPred [7] | MoTIF [8] | Refexp [9] | VWB AG [10] | VWB EG [10] | AndroidWorld [11] | Average   |
> | :----------------------------------- | :--------------- | :------------------ | :------------- | :---------- | :----------- | :------------ | :------------ | :------------------ | :---------- |
> | Qwen2.5-VL-7B-Base (paper report)  | 84.70          | -                 | -            | -         | -          | -           | -           | -                 | -         |
> | Qwen2.5-VL-7B-Base (our report)    | 84.01          | 80.34             | 48.25        | 71.93     | 79.46      | 72.81       | 90.07       | 10.34             | 67.15     |
> | Qwen2.5-VL-7B-Continue-Train       | 84.91          | 84.43             | 59.50        | 68.30     | 72.13      | 67.96       | 93.70       | 12.06             | 67.87     |
> | Qwen2.5-VL-7B-SFT-Continue-Train   | 85.06          | 85.06             | 59.30        | **80.47** | **83.19**  | 68.93       | 92.01       | 12.93             | 70.87     |
> | Qwen2.5-VL-7B-ST-RL-Continue-Train | 85.53          | **85.61**         | **60.45**    | 79.41     | 77.88      | 70.87       | **92.98**   | 14.65             | 70.92     |
> | Qwen2.5-VL-7B-MT-RL-Continue-Train | **86.08**      | **85.61**         | 59.40        | 78.18     | 81.77      | **76.70**   | **92.98**   | **15.51**         | **72.03** |
>
> 3. **Weakness 3 Originality:** While GE-Lab is novel, the concept of synthetic navigation environments echoes prior work (e.g., text-based games), and the conclusions (e.g., RL enhances exploration) are not entirely new, reducing the paper’s distinctiveness.
> - Response:
>   * Although the concept of synthetic navigation environments is not new, we believe that the contribution of this study is distinct in a number of important ways.
> First, prior synthetic navigation environments have largely been text-based. In contrast, this study is the first to construct a GUI simulation environment that supports interactive training and provides accurate reward signals. **This simulated environment enables us to validate the effectiveness of reinforcement learning methods**, providing a foundation for future real-world environment development and enhancing agent performance in real-world settings.
>   * Second, while it may seem intuitive that reinforcement learning can enhance the exploration capabilities of agents, **there are still several practical challenges, such as reward hacking in RL and the lack of progression during MT-RL training**. To address these challenges, we mitigated reward hacking by adjusting the training data proportions and improved the efficiency and stability of MT-RL through curriculum learning. The insights gained from addressing these issues in GE-Lab can be effectively transferred to real-world applications.
>
> 4. **Weakness 4 Clarity:** Although generally clear, the paper’s failure to address how its simplified model applies to complex GUI tasks may confuse readers expecting real-world relevance.
> - Response:
>   * We further investigated the generalizability of our approach by incorporating SFT on a subset of real-world data. Specifically, we augmented our training pipeline (SFT, ST-RL, MT-RL) with 24k samples drawn from publicly available datasets, including AITW [1], AITZ [2], AMEX [3], and Mind2Web [4]. We then evaluated the resulting agents on both real-world grounding and interactive benchmarks.
>   * The results demonstrate that **our training methodology yields agents with strong generalization capabilities across both simulated and real-world GUI tasks.** Notably, MT-RL outperforms ST-RL, which in turn outperforms SFT, in terms of accuracy. This observation mirrors the trend observed in our simulated environment as reported in the main text and reinforces our key findings: (1) MT-RL yields better generalization than ST-RL, and (2) ST-RL generalizes better than SFT.
>   * We opt to conduct the experiments in simulation primarily due to the significant development cost associated with providing accurate reward signals in real-world environments. **Simulation enables us to validate the proposed multi-turn reinforcement learning framework in a controlled manner.** This not only demonstrates the feasibility of our approach but also lays a foundation for future development of real-world environments, which would ultimately advance the capabilities of GUI agents.
>
> 5. **Suggested References:**
>
> * We will carefully review the suggested references and incorporate them to enhance our writing and references section.
>
> **Reference** (Due to space limitations, our reference list includes only the titles of the articles. For reference [9-11], please refer to our response to reviewer DUjd.)
>
> [1] Androidinthewild: A large-scale dataset for android device control[J].
>
> [2] Android in the zoo: Chain-of-action-thought for GUI agents[J].
>
> [3] Amex: Android multi-annotation expo dataset for mobile GUI agents[J].
>
> [4] Mind2web: Towards a generalist agent for the web[J].
>
> [5] Seeclick: Harnessing GUI grounding for advanced visual GUI agents[J].
>
> [6] Os-atlas: A foundation action model for generalist GUI agents[J].
>
> [7] Autogui: Scaling GUI grounding with automatic functionality annotations from llms[J].
>
> [8] A dataset for interactive vision-language navigation with unknown command feasibility[C]

---

> > ### Comment · Reviewer_VsGX · 2025-08-05
> >
> > Thank you for the additional experiments, which partially address my concerns about transfer learning and practical impact. The results in Tables 1 and 2 do demonstrate some effectiveness of GE-Lab for real-world GUI tasks, and I will adjust my score accordingly.
> >
> > However, these extensive experiments (covering multiple datasets, new action spaces, and comprehensive benchmarks) were not included in the original paper, making the current presentation quite disorganized.
> >
> > **Request:** Please properly integrate these experimental results and methodology into a revised version of the manuscript. The paper would benefit from:
> > - Including the real-world transfer experiments in the main evaluation section
> > - Providing detailed methodology for the AITW/AITZ/AMEX/Mind2Web experiments
> > - Better organization of the experimental narrative
> >
> > The additional evidence is valuable, but it needs to be properly structured within the paper to be fully convincing.

---

> > > ### Author Response · Authors · 2025-08-05
> > > **Response to Reviewer VsGX**
> > >
> > > * Response
> > >   * We appreciate that Reviewer VsGX found the additional experiments effectively addressed potential concerns about the practical applicability of our approach to real-world scenarios.
> > >   * In the revised manuscript, a new Section 6 will be added to provide a evaluation of our approach in real-world scenarios, and the content will be carefully restructured to **clearly and logically present the methodology and results of our approach on real-world GUI benchmarks**. Accordingly, the original Conclusion section will be renumbered as Section 7. Specifically, the new Section 6 will be organized into three main parts:
> > >       * Section 6.1: Experimental Setup – covering datasets, model training configurations, and evaluation protocols.
> > >       * Section 6.2: Quantitative Analysis – presenting numerical results and analysis across different training paradigms.
> > >       * Section 6.3: Qualitative Analysis – discussing case studies and practical implications.
> > >   * The detailed contents are as follows:
> > >       * 6.1 **Experimental Setup**
> > >          * 6.1.1 **Real-World Datasets for Continue Training**
> > >        We will provide detailed descriptions of the datasets used during the continue training phase, including AITW [1], AITZ [2], AMEX [3], and Mind2Web [4]. To ensure transparency and reproducibility, we will also release the sampled data used in our experiments.
> > >          * 6.1.2 **Training Details of Continue-Trained Models**
> > >             The following models will be documented with complete training configurations: Qwen2.5-VL-7B-Continue-Train, Qwen2.5-VL-7B-SFT-Continue-Train, Qwen2.5-VL-7B-ST-RL-Continue-Train, Qwen2.5-VL-7B-MT-RL-Continue-Train. We will specify training objectives, optimization schedules, and other implementation details necessary to reproduce these models.
> > >          * 6.1.3 **Real-World Benchmark Descriptions and Evaluation Setup**
> > >             We will provide clear introductions to the real-world benchmarks used for evaluation. This includes a description of the task settings, action space definition, and evaluation metrics, offering the necessary context for understanding generalization performance across domains.
> > >       * 6.2 **Quantitative Analysis of Experimental Results**
> > >        The results previously presented in Tables 1 and 2 of the rebuttal will be incorporated into the main manuscript. We will include a detailed quantitative analysis to clarify how different training paradigms (SFT, ST-RL, MT-RL) influence generalization, robustness, and transferability across tasks.
> > >       * 6.3 **Qualitative Analysis and Case Studies**
> > >        To further highlight the strengths and limitations of the current approaches, we will include representative case studies comparing the behavior of SFT, ST-RL, and MT-RL methods on real-world GUI tasks. These examples will ground our discussion on practical deployment, and we will reflect on current bottlenecks and potential pathways toward more effective real-world transfer.
> > >   * We believe these revisions will significantly improve the coherence and completeness of the paper and make the practical contributions of our work more convincing. We sincerely thank the reviewer VsGX again for the constructive feedback, which plays a crucial role in enhancing the quality of our work.

---

### Official Review · Reviewer_htJ2 · 2025-07-03

**Clarity:** 3
**Significance:** 3
**Originality:** 2
**Rating:** 5
**Confidence:** 4

**Summary:**

This paper introduces GE-Lab, an environment that composes navigation apps to train and test GUI agents. On this environment, they tested three types of training: supervised fine-tuning, single-step reinforcement learning and multi-turn reinforcement learning. Overall, they show how supervised fine-tuning is effective for in-distribution data, while single-step reinforcement learning is effective for out-of-distribution data. Finally, multi-turn reinforcement learning usually outperforms the previous strategies, especially in the interactive environment.

**Questions:**

- Can you show the difference in training time for the three methods?
- Can you show the sampling efficiency difference for the three methods?
- To improve the clarity, introduce the acronym before using them.

**Ethical Concerns:**

["NO or VERY MINOR ethics concerns only"]

**Final Justification:**

I have raised my score, as they also assess the time efficiency of the 3 methods in the appendix.
I recommend acceptance, as this work is valuable, although it doesn't present surprising results, it scientifically assesses the benefit in multi-turn RL.

**Limitations:**

yes

**Paper Formatting Concerns:**

Emoji in the title

**Quality:**

3

**Strengths And Weaknesses:**

The paper is well written, and the set of experiments is clear and informative. The paper is significant, although the environment is too simple for a real-case scenario. The work is original, even if prior works explored the use of multi-turn RL for agents. The experiments showed multi-turn RL generally improves performances; however they lack to analyze the efficiency in term of training speed and sample efficiency.

---

> ### Author Rebuttal · Authors · 2025-07-31
>
> We would like to thank the reviewer for their positive and constructive feedback. **We appreciate the recognition of the clarity and significance of our experimental setup, as well as the acknowledgment of the originality of our work in the context of multi-turn RL for GUI agents.** We also value the reviewer’s suggestion regarding the analysis of training speed and sample efficiency, and we have now included additional details on this aspect in the following sections of the revised manuscript. The feedback has been invaluable in helping us enhance the comprehensiveness of our work.
> Below are our detailed responses to the Reviewer htJ2.
>
> 1. **Question 1:** Can you show the difference in training time for the three methods?
> - Response: The difference in training time for the three methods.
>   * As mentioned in Appendix A.1 (line 478-482), A typical training time for a single SFT agent is approximately 3 to 4 GPU hours. The training of ST-RL agent generally requires about 48 GPU hours, while MT-RL agent typically trains for approximately 36 GPU hours.
>   * As stated in Appendix A.1, Dataset & Benchmark, the SFT dataset consists of 60,864 samples, the ST-RL dataset contains 12,439 samples, and the MT-RL dataset is based on online interactive training with 2,162 tasks. **The average training time per sample/task is shown in the Table 1 below.**
>
> **Table 1:** Training Time and Sampling Efficiency Comparison for the Three Methods
>
> | Method | Dataset Size  | Training Time | Average Training Time              | Interaction Count | Feedback                               |
> | -------- | --------------- | --------------- | ------------------------------------ | ------------------- | ---------------------------------------- |
> | SFT    | 60864 samples | 3~4 GPU Hours | 0.000049~0.000066 GPU Hours/sample | 60864             | Manually labeled                       |
> | ST-RL  | 12439 samples | 48 GPU hours  | 0.00386 GPU Hours/sample           | 99512             | Manually labeled      |
> | MT-RL  | 2162 tasks    | 36 GPU hours  | 0.0167 GPU Hours/task              | ~89939            | Environment Feedback |
>
> 2. **Question 2:** Can you show the sampling efficiency difference for the three methods?
> - Response: The sampling efficiency difference for the three methods.
>   * For both ST-RL and MT-RL, the number of generations per round is 8. **Table 1 highlights significant differences in sampling efficiency across the three methods.** SFT has the lowest average training time per sample, indicating high efficiency for in-distribution data. In contrast, ST-RL and MT-RL show higher training times per sample or task, reflecting the additional time consumption required by reinforcement learning. However, ST-RL and MT-RL achieve higher interaction counts, suggesting that they are more effective in exploring diverse states and generating robust agent behaviors, particularly in out-of-distribution and interactive settings.
>
> 3. **Question 3:** To improve the clarity, introduce the acronym before using them.
> - Response: About acronym.
>   * We will carefully review the acronyms in the final version to avoid any potential confusion for the readers.

---

> > ### Author Response · Authors · 2025-08-08
> > **Response to Reviewer htJ2**
> >
> > Dear Reviewer htJ2,
> >
> > Thank you very much for your constructive comments and for raising further concerns regarding training time and sampling efficiency. We have carefully addressed these in our latest response, and we would greatly appreciate it if you could take a moment to review it.
> >
> > As the discussion phase is nearing its deadline, your timely feedback would be invaluable in helping us further refine and improve our work. We sincerely welcome any additional thoughts you may have — your insights are highly valuable to us.
> >
> > Best regards,
> >
> > The authors

---

> > > ### Comment · Reviewer_htJ2 · 2025-08-08
> > >
> > > Dear Authors,
> > >
> > > Thank you for highlighting the results in the appendix. I apologize for the delayed response.
> > >
> > > The results you presented are insightful. I am surprised to see that the training time for the multi-turn RL is lower than that for the single-step RL. Do you have any insights on this? Does this measurement include the time taken by the environment? It would also be valuable to assess the environment's speed in relation to the training time, as I suspect that your simplified environment is much more efficient than what a typical GUI application would be.
> > >
> > > In any case, I find your work valuable, so I am raising my score to recommend acceptance.
> > >
> > > Thank you.

---

> > > > ### Author Response · Authors · 2025-08-08
> > > > **Response to Reviewer htJ2**
> > > >
> > > > We sincerely thank the reviewer htJ2 for the continued recognition of our work and for the constructive comments, which are highly valuable for improving our paper.
> > > >
> > > > Regarding the question on training time, we confirm that the reported measurements include the time taken by the environment. In our simulation setup, feedback for each interaction is generated by checking the page ID of the current screen, the click position predicted by the agent, and the environment’s meta information, and then returning the updated page ID after the operation. For example, if the current state is page_6 and the predicted click falls within the bounding box of the Home icon (according to the cached meta information), the environment transitions to page_0. This process does not require real-time rendering — all screens are pre-rendered and cached — and therefore only incurs millisecond-level latency. Compared with the online rollout time of the model (~13 s/sample), this overhead is negligible. We agree that this simulation environment offers a clear advantage over typical GUI applications with respect to interaction latency.
> > > >
> > > > The reason why the training time for multi-turn RL (MT-RL) is lower than that for single-step RL (ST-RL) is that the MT-RL agent interacts with the environment fewer times than the cumulative single-step inferences of the ST-RL agent. As shown in Table 1 of the rebuttal (“Training Time and Sampling Efficiency Comparison for the Three Methods”), ST-RL has an interaction count of 99,512, whereas MT-RL has approximately 89,939. In practice, we selected the final models by monitoring the reward curves of ST-RL and MT-RL during training, and MT-RL reached higher performance with fewer interactions.
> > > >
> > > > We will incorporate these details from the rebuttal into the revised manuscript to improve clarity. Once again, we thank the reviewer htJ2 for the careful assessment and for the positive discussion of our work.

---

### Official Review · Reviewer_DUjd · 2025-07-03

**Clarity:** 2
**Significance:** 3
**Originality:** 3
**Rating:** 4
**Confidence:** 3

**Summary:**

This paper introduces GUI Exploration Lab (GE-Lab), a simulation environment for training and evaluating GUI navigation agents with reinforcement learning. The environment enables flexible compositionality of screens, icons, and navigation graphs, allowing researchers to generate large-scale, customizable benchmarks and detailed environment exposure for agents. Through extensive experiments, the authors benchmark three training regimes (SFT, single-turn RL, and multi-turn RL) using the new environment, showing that SFT helps agents memorize, ST-RL improves generalization, and MT-RL promotes active exploration and interactive recovery. The manuscript includes quantitative and qualitative analysis of these regimes’ performances under in-distribution, out-of-distribution, and interactive tasks, and investigates reward design and pretraining regimen effects.

**Questions:**

Overall, I prefer to be borderline; but, since a strictly neutral rating is not available, I lean slightly positive and therefore give a borderline accept. I will make my final decision after reading the authors’ rebuttal.

**Ethical Concerns:**

["NO or VERY MINOR ethics concerns only"]

**Final Justification:**

The authors have addressed the main concerns raised in the review. Their additional experiments on real-world GUI benchmarks and further clarification of GE-Lab’s novelty provide support for the paper’s contributions. Initially, I felt the work was borderline but leaned toward a borderline accept due to the lack of a neutral rating option. With the clarification provided, I now believe the current score is appropriate and will maintain my positive rating.

**Limitations:**

yes

**Paper Formatting Concerns:**

no big issues

**Quality:**

3

**Strengths And Weaknesses:**

Strengths

* GE-Lab is a meaningful addition to the field, providing a flexible simulation platform capable of modeling complex GUI structures, navigation graphs, and icon layouts, with controllable OOD scenarios for systematic evaluation.
* The experimental setup and analysis are thorough, covering in-distribution, OOD, and interactive settings, with clear ablation studies and insightful discussions.
* The paper is generally well-written and easy to follow, with clear figures and explanations.

Weaknesses
* Limited environment realism: Despite its flexibility, the simulated environment is essentially tree-structured and features a simplified action space that lacks the diversity and complexity of real-world GUIs. This limitation, which the authors themselves acknowledge, may reduce the generalizability of the findings and constrains the practical implications for deployed agents.
* Missing real-world validation: Although OOD generalization is evaluated within the simulation (Table 3), the paper does not test the transferability of the trained agents to real-world GUIs or public benchmarks (e.g., Mind2Web). It remains unclear how the learned policies and training dynamics would translate outside the simulated setting.
* Incremental novelty of simulation-based training: The idea of using simulation platforms to train and benchmark GUI agents, particularly in reinforcement learning settings, is not entirely novel, with existing platforms such as AndroidWorld already exploring similar directions. The paper would benefit from a clearer positioning relative to prior work in this area.

---

> ### Author Rebuttal · Authors · 2025-07-31
>
> We would like to thank the reviewer for their thoughtful feedback and positive evaluation of our work. We appreciate the recognition of GE-Lab as a valuable contribution to the field, particularly for its **flexibility in modeling complex GUI structures and controllable OOD scenarios.** We also thank the reviewer for **highlighting the thoroughness of our experimental setup and the clarity of our ablation studies**, which helped us assess the performance of different training regimes. The feedback encourages us to further refine our work and explore additional avenues for improving the generalization and interactivity of GUI navigation agents.
> Below are our detailed responses to the Reviewer DUjd.
>
> 1. **Weakness 1:** Limited environment realism. Whether the simplified simulation environment and action space reduce the generalizability of the findings and constrains the practical implications for deployed agents.
>   - Response:
>     * We acknowledge that the simulated environment used in our study exhibits limited diversity and complexity compared to real-world GUIs. However, we would like to emphasize that evaluating performance on real-world benchmarks is not the primary focus of this work. Instead, our goal is to validate the feasibility of using reinforcement learning for screen navigation tasks. By further adopting real-world icons and layouts inspired by actual apps within GE-Lab, the domain gap can be largely reduced.
>     * In fact, the agent trained in the simulated environment demonstrates a notable degree of generalization to real-world GUIs. To assess this, we manually constructed a static benchmark by sampling 1,569 instances from several open-source real-world GUI datasets, including AITW [1], AITZ [2], AMEX [3], and Mind2Web [4]. The test samples cover a broader action space: CLICK, COMPLETE, WAIT, SCROLL, and TYPE. Without any further fine-tuning, we evaluated our agent, which had not encountered WAIT, SCROLL, or TYPE actions during the training process, directly on this benchmark.
>      **The results are presented in the Table 1 below. Despite lacking explicit training on WAIT, SCROLL, and TYPE, the agent still achieves non-trivial accuracy on these actions.** This suggests that the agent possesses some degree of atomic competence in handling GUI elements beyond its training distribution, highlighting the generalizability of our approach and its practical implications for real-world deployment.
>
> **Table 1:** Performance of the agent on real-world static benchmark with extended action space.
>
> | Action   | Total Number | Success Number | Success Rate |
> | ---------- | -------------- | ---------------- | -------------- |
> | CLICK    | 971          | 622            | 64.06%       |
> | COMPLETE | 230          | 221            | 96.09%       |
> | WAIT     | 231          | 99             | 42.86%       |
> | TYPE     | 129          | 119            | 92.25%       |
> | SCROLL   | 8            | 5              | 62.50%       |
>
> 2. **Weakness 2:** Missing real-world validation.  Although OOD generalization is evaluated within the simulation (Table 3 in paper), the paper does not test the transferability of the trained agents to real-world GUIs or public benchmarks.
>  - Response:
>     - We further investigated the generalizability of our approach by incorporating SFT on a subset of real-world data. Specifically, we continue training our agents (SFT, ST-RL, MT-RL) with 24k samples drawn from publicly available datasets, including AITW [1], AITZ [2], AMEX [3], and Mind2Web [4]. We then evaluated the resulting agents on both real-world grounding and interactive benchmarks. The results support the claim that our training framework yields agents with strong generalization capabilities across both simulated and real-world GUI tasks.
>     - **As shown in Table 2 below, the results demonstrate that our training framework yields agents with strong generalization capabilities across both simulated and real-world GUI tasks.** Notably, MT-RL outperforms ST-RL, which in turn outperforms SFT, in terms of accuracy. This observation mirrors the trend observed in our simulated environment as reported in the main text and reinforces our key findings: (1) MT-RL training yields better generalization than ST-RL training, and (2) ST-RL generalizes better than SFT. Additionally，we appreciate review's recommendation to test on the Mind2Web. While Mind2Web is a comprehensive bench for developing and evaluating web agents, its original version is primarily text-based, focusing on HTML and textual descriptions of tasks. Although a multimodal version of Mind2Web, which pairs HTML documents with webpage screenshots, has been released, its core structure and tasks are still fundamentally designed for web agents that operate on textual data. Visual perception of the GUI is a primary input channel in our agent, Therefore, the Mind2Web benchmark does not align with the architectural and data requirements of our multimodal GUI agent.
>     - We opt to conduct the experiments in simulation primarily due to the significant development cost associated with providing accurate reward signals in real-world environments. Simulation enables us to validate the proposed MT-RL framework in a controlled manner. This not only demonstrates the feasibility of our approach but also lays a foundation for future development of real-world environments, which would ultimately advance the capabilities of GUI agents.
>
> **Table 2:** Performance of the agent on real-world grounding benchmark and interactive benchmark.
>
> | Model                              | ScreenSpot [5] | ScreenSpot-v2 [6] | FuncPred [7] | MoTIF [8] | Refexp [9] | VWB AG [10] | VWB EG [10] | AndroidWorld [11] | Average   |
> | :----------------------------------- | :--------------- | :------------------ | :------------- | :---------- | :----------- | :------------ | :------------ | :------------------ | :---------- |
> | Qwen2.5-VL-7B-Base (paper report)  | 84.70          | -                 | -            | -         | -          | -           | -           | -                 | -         |
> | Qwen2.5-VL-7B-Base (our report)    | 84.01          | 80.34             | 48.25        | 71.93     | 79.46      | 72.81       | 90.07       | 10.34             | 67.15     |
> | Qwen2.5-VL-7B-Continue-Train       | 84.91          | 84.43             | 59.50        | 68.30     | 72.13      | 67.96       | 93.70       | 12.06             | 67.87     |
> | Qwen2.5-VL-7B-SFT-Continue-Train   | 85.06          | 85.06             | 59.30        | **80.47** | **83.19**  | 68.93       | 92.01       | 12.93             | 70.87     |
> | Qwen2.5-VL-7B-ST-RL-Continue-Train | 85.53          | **85.61**         | **60.45**    | 79.41     | 77.88      | 70.87       | **92.98**   | 14.65             | 70.92     |
> | Qwen2.5-VL-7B-MT-RL-Continue-Train | **86.08**      | **85.61**         | 59.40        | 78.18     | 81.77      | **76.70**   | **92.98**   | **15.51**         | **72.03** |
>
> 3. **Weakness 3:** Incremental novelty of simulation-based training. The paper would benefit from a clearer positioning relative to prior work in this area.
> - Response:
>   * We have carefully surveyed several existing GUI-based environments—AndroidWorld (116 tasks), OSWORLD (369 tasks), and WebArena (812 tasks)—which target phone-use, computer-use, and browser-use scenarios, respectively. While these platforms serve as valuable benchmarks, the number of tasks in each remains relatively limited. This limitation stems from the significant engineering effort involved in designing tasks that support automated and accurate reward computation — often necessitating task-specific setups such as generating predefined files, which makes large-scale, diverse and scalable training infeasible. **To the best of our knowledge, no existing work has demonstrated multi-turn reinforcement learning training in real-world GUI environments, highlighting the practical challenges and the novelty of our simulation-driven approach.**
>   * In contrast to the above three platforms, GE-Lab, as a simulated environment, is capable of generating an unlimited number of tasks while also automating the provision of accurate reward signals. This is critical for scaling up multi-turn RL training and for exploring learning dynamics that are otherwise infeasible to study in constrained real-world environments.
>
> **Reference** (Due to space limitations, our reference list includes only the titles of the articles.)
>
> [1] Androidinthewild: A large-scale dataset for android device control[J].
>
> [2] Android in the zoo: Chain-of-action-thought for GUI agents[J].
>
> [3] Amex: Android multi-annotation expo dataset for mobile GUI agents[J].
>
> [4] Mind2web: Towards a generalist agent for the web[J].
>
> [5] Seeclick: Harnessing GUI grounding for advanced visual GUI agents[J].
>
> [6] Os-atlas: A foundation action model for generalist GUI agents[J].
>
> [7] Autogui: Scaling GUI grounding with automatic functionality annotations from llms[J].
>
> [8] A dataset for interactive vision-language navigation with unknown command feasibility[C]
>
> [9] Uibert: Learning generic multimodal representations for ui understanding[J].
>
> [10] Visualwebbench: How far have multimodal llms evolved in web page understanding and grounding?[J].
>
> [11] Androidworld: A dynamic benchmarking environment for autonomous agents[J].

---

> > ### Comment · Reviewer_DUjd · 2025-08-07
> > **Thanks for the rebuttal**
> >
> > I thank the reviewer for the clarifications, which largely address my initial concerns. Initially, I felt the work was borderline but leaned toward a borderline accept due to the lack of a neutral rating option. With the clarification provided, I now believe the current score is appropriate and will maintain my positive rating.

---

> > > ### Author Response · Authors · 2025-08-08
> > > **Response to Reviewer DUjd**
> > >
> > > We sincerely thank the reviewer DUjd for the thorough assessment of our paper and rebuttal, and for the constructive feedback that has helped us better articulate our contributions. We also appreciate the reviewer’s recognition of the value of our work.

---

### Decision · Program_Chairs · 2025-09-17

**Decision:**

Accept (poster)

**Comment:**

This paper introduces GUI Exploration Lab, a simulation environment engine for GUI agents. This environment enables flexible GUI compositions with screens, icons, and navigation graphs. This not only allows scale-scale GUI synthesis but also creates controllable OOD evaluation. Experiments offer a systematic comparison of different training paradigms, including SFT, SF-RL, and MT-RL.

During the rebuttal, the authors provided a real-world evaluation and training efficiency analysis, which further strengthened the work. I would encourage the authors to include the new results and analysis in the final version.